

# Spatial distribution of environmental indicators in surface sediments of Lake Bolshoe Toko, Yakutia, Russia

Boris K. Biskaborn[1*], Larisa Nazarova[1,2,3], Lyudmila A. Pestryakova[4], Liudmila Syrykh[5], Kim Funck[1,6], Hanno Meyer[1], Bernhard Chapligin[1], Stuart Vyse[1], Ruslan Gorodnichev[4], Evgenii Zakharov[4,7], Rong Wang[8], Georg Schwamborn[1,9], Bernhard Diekmann[1,2]

*Corresponding author's Email: boris.biskaborn@awi.de

*1 Alfred Wegener Institute Helmholtz Centre for Polar and Marine Research, Potsdam, Germany*
*2 University of Potsdam, Potsdam, Germany*
*3 Kazan Federal University*
*4 North-Eastern Federal University of Yakutsk, Russia*
*5 Herzen State Pedagogical University of Russia, St. Petersburg, Russia*
*6 Humboldt University Berlin, Germany*
*7 Institute for Biological Problems of Cryolithozone Siberian Branch of RAS, Yakutsk, Russia*
*8 Key Laboratory of Submarine Geosciences, State Oceanic Administration, Hangzhou, China*
*9 Free University of Berlin, Berlin, Germany*

**Manuscript status**:
Approved by all authors. English proofread.

**Keywords:**
Diatoms, chironomids, XRF elements, XRD minerals, grain-size distribution, oxygen isotopes, organic carbon

## Abstract

Rapidly changing climate in the northern hemisphere and associated socio-economic impacts require reliable understanding of lake systems as important freshwater resources and sensitive sentinels of environmental changes. To better understand time-series data in lake sediment cores it is necessary to gain information on within-lake spatial variabilities of environmental indicator data. Therefore, we retrieved a set of 38 samples from the sediment surface along spatial habitat gradients in the boreal, deep, and yet pristine Lake Bolshoe Toko in southern Yakutia, Russia. Our methods comprise laboratory analyses of the sediments for multiple proxy parameters including diatom and chironomid taxonomy, oxygen isotopes from diatom silica, grain size distributions, elemental compositions (XRF), organic carbon contents, and mineralogy (XRD). We analysed the lake water for cations, anions and isotopes. Our results show that the diatom assemblages are strongly influenced by water depth and dominated by planktonic species, i.e.





*Pliocaenicus bolshetokoensis*. Species richness and diversity is higher in the
northern part of the lake basin, associated with the availability of benthic, i.e.
periphytic, niches in shallower waters. $\delta^{18}O_{diatom}$ values are higher in the deeper
south-western part of the lake probably related to water temperature differences.
The highest amount of the chironomid taxa underrepresented in the training set
used for palaeoclimate inference was found close to the Utuk river and at southern
littoral and profundal sites. Abiotic sediment components are not symmetrically
distributed in the lake basin but vary along restricted areas of differential
environmental forcings. Grain size and organic matter is mainly controlled by both,
river input and water depth. Mineral (XRD) data distributions are influenced by the
methamorphic lithology of the Stanovoy mountain range, while elements (XRF) are
intermingled due to catchment and diagenetic differences. We conclude that the
lake represents a suitable system for multiproxy environmental reconstruction based
on diatoms (including oxygen isotopes), chironomids and sediment-geochemical
parameters.

## 1   Introduction

Over the past few decades, the atmosphere in boreal and high elevation regions
has warmed faster than anywhere else on Earth (Pepin et al., 2015;Huang et al.,
2017). Dramatic socio-economic and ecological consequences are expected
(AMAP, 2017) as well as substantial feedbacks from thawing permafrost and
associated release of greenhouse gas in the global climate system (Schuur et al.,
2015). Boreal Russia, as compared to the rest of the world, has been reported as a
hot-spot region, where air temperature increases lead to substantial ground
warming over the last decade (Biskaborn et al., 2019). Estimations of the accurate
amplitude of environmental impact suffer from imprecise understanding of ecological
indicators of past environmental conditions (Miller et al., 2010). Lake ecosystems,
whose development is archived in their sediments, act as sensitive sentinels of
environmental changes (Adrian et al., 2009), but rely on careful interpretation of
suitable proxy data. Proxy information on present and past ecological conditions is
provided by various biological and physicochemical properties of the sediment
components (Meyer et al., 2015;Solovieva et al., 2015;Nazarova et al., 2017a).
However, the spatial within-lake distributions of preserved remnants of ecosystem
inhabitants and associated sediment-geochemical properties, depend on habitat
differences between the epilimnion and the hypolimnion (Raposeiro et al., 2018),
and are therefore expected to be non-uniform. Accordingly, precise
paleolimnological reconstruction of past environmental variability requires a
profound understanding of the recent within-lake heterogeneity.



Our approach comprises commonly applied sedimentological variables that help
to gain a holistic view on a lake's depositional history, including diatom and
chironomid taxonomy, $\delta^{18}O_{diatom}$, grain size distributions, elemental compositions,
organic carbon contents, and mineralogy. Abiotic sediment preferences represent
signals that result from external input of material and lake-internal conditions during
deposition as well as post-sedimentary diagenetic processes near the sediment
surface (Biskaborn et al., 2013b;Bouchard et al., 2016). To reliably identify true
environmental signals, it is therefore necessary to apply multiproxy approaches that
enable an understanding of lake-internal filters between original external forcing and
the resulting preferences of the sediment deposition (Cohen, 2003).
Diatoms (unicellular, siliceous microalgae) represent a major part of the aquatic
primary producers. They appear ubiquitous and their opaline frustules are well
preserved in the sedimentary record, allowing exact identification down to sub-
species level by high-resolution light microscope analysis (Battarbee et al., 2001).
Diatoms are among the most applied bioindicators for past and present ecosystem
changes in boreal environments (Miller et al., 2010;Pestryakova et al., 2012;Hoff et
al., 2015;Herzschuh et al., 2013;Biskaborn et al., 2012;Biskaborn et al.,
2016;Palagushkina et al., 2017;Douglas and Smol, 2010). Widespread responses of
planktonic diatoms to recent climate change prove that lakes in the northern
hemisphere often have already crossed important ecological thresholds (Smol and
Douglas, 2007;Rühland et al., 2008). The very rapid life cycles of the specimen of
days to weeks (Round et al., 1990) enables changes in diatom assemblages on
very short time-scales in response to changes in environmental circumstances, e.g.
cooling or warming (Anderson, 1990). The link between climate change and
diatoms, however, cannot easily be addressed via simple temperature-inference
models. The situation demands a more complete understanding of the interactions
between the aquatic ecosystem with lake habitat preferences, hydrodynamics and
catchment properties (Anderson, 2000;Palagushkina et al., 2012;Biskaborn et al.,
2016;Bracht-Flyr and Fritz, 2012;Hoff et al., 2015). It is thus necessary to identify
the relationship between diatom species occurrence, the isotopic composition of
their opaline valves, and internal physico-limnological factors (Heinecke et al., 2017)
within spatial heterogenic lake systems before drawing direct inferences about
external climatic driven factors from single core studies.
Chironomid larvae (Insecta: Diptera) can make up to 90% of the aquatic
secondary production (Herren et al., 2017;Nazarova et al., 2004) and hence their
preserved head capsules represent well the aquatic heterotrophic bottom-dwelling
ecosystem component (Nazarova et al., 2008;Syrykh et al., 2017;Brooks et al.,
2007). Furthermore, literature reports a net mutualism of chironomids and benthic
algae between the primary consumer and primary producer trophic levels in benthic
ecosystems (Specziar et al., 2018;Zinchenko et al., 2014). Factors influencing the



spatial distribution of chironomids within single lakes are water temperature
(Nazarova et al., 2011;Luoto and Ojala, 2018), sedimentological habitat
characteristics (Heling et al., 2018) and/or water depth and nutrients (Yang et al.,
2017), as well as hypolimnetic oxygen (Stief et al., 2005) and the availability of
water plants (Raposeiro et al., 2018;Wang et al., 2012b).
Secondary factors influencing the spatial distribution of subfossil assemblages
are selective transitions from living communities to accumulation of dead remains.
Both, biological remains and physico-chemical properties are influenced by
sediment resuspension and redistribution processes described as sediment
focusing (Hilton et al., 1986) which mainly depend on slope steepness (Hakanson,
1977) or, in shallow areas, wind-induced bottom shear stress (Bennion et al.,
2010;Yang et al., 2009). Nevertheless, it already has been proven for other lake
sites that within-lake bioindicator distributions are laterally non-uniform, contradicting
the assumption that mixing processes cause homogenous microfossil assemblages
before deposition (Anderson, 1990;Wolfe, 1996;Anderson et al., 1994;Earle et al.,
1988;Kingston et al., 1983;Puusepp and Punning, 2011;Stewart and Lamoureux,
2012;Yang et al., 2009). However, many palaeolimnological studies have hitherto
ignored that single-site approaches using only one sediment core do not encompass
the full spatial extent and natural variability of the entire lake sediment archive.
Heggen et al. (2012) reported that sediment cores from the deep centre of small and
shallow lakes with high spatial proxy variability in the littoral zones contain
representative bioindicator assemblages. The authors also conclude, that in larger
and deeper lakes similar multi-site studies are necessary to make recommendations
about the "ideal" coring positions for multi-proxy palaeolimnological studies.
In this respect, our general research question was: how spatially reliable are
palaeolimnological proxy data in a complex lake system? To answer this question,
we set up our research hypotheses: (1) Bioindicators will respond to different habitat
properties and hence vary spatially in a complex lake system. (2) Water depth and
sediment-geochemical parameters will correlate with species assemblages at
different locations within a lake basin.
An analysis of spatio-temporal within-lake bioindicator distribution requires a
suitable and large lake system with an anthropogenically untouched ecosystem and
sufficient variability in water depth, catchment setting, and the sedimentological
regime. These demands are met by Lake Bolshoe Toko which was considered as
the deepest lake in Yakutia (Zhirkov et al., 2016), located in the Sakha Republic,
Russia (Fig. 1). Our study aims to gain a better local understanding of proxy data for
planned palaeoenvironmental analyses of long sediment cores from Bolshoe Toko.
Therefore, our objectives are (1) to detect the spatial variability of abiotic (elements,
minerals, grain size) and biotic (diatoms, chironomids, organic carbon) components
of the lake's surface sediments, (2) to reveal the causal relationship between the

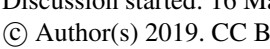

distribution of aquatic microfossils, lake basin features, and sedimentary
parameters, and (3) to attribute proxy variability to stressors and factors of the lake
basin and its catchment.

## 2 Study site

The maximum diameter of Lake Bolshoe Toko (56°15'N, 130°30'E, 903 m.a.s.l) is
15.4 km, the maximum width is 7.6 km, the maximum water depth is 78 m (average
30.5 m), the surface area 8500 Ha, and the water transparency is 9.8 m and the
lake was indexed as "clean oligotrophic water" (Zhirkov et al., 2016). The north
eastern lake basin is shallower (<30 m) compared to the south western part of the
lake (up to 80 m). The Utuk river runs through Lake Maloe Toko and brings water
from the southern igneous catchment. The Lake Maloe Toko (called "small Toko",
size 2.7 x 0.9 km, 168 m depth, tectonic origin) is located between high mountains
south of Bolshoe Toko. The river inflow south of Bolshoe Toko forms deltaic
sediments. The bay in the southeast is called Zaliv Rybachiy ("Fishing bay"). It is
partly separated from the main basin and supplied with water by a small creek that
itself is connected to a small lake (Fig. 1). The bay is reported to have a somewhat
different fauna as compared to the Bolshoe Toko main basin, i.e. occurrence of fish
that is typical for small lakes and not found out of the basin (Semenov, 2018). The
"Banya lake" in the northeast is completely separated from Bolshoe Toko and was
hence not considered in this study. The Mulam river is the lake's predominant
outflow towards the northern direction along the south eastern border of Yakutia
flowing into the Uchur, Aldan and finally into the Lena rivers.
There are no permanent settlements in the study area. During the time of field
work there was a temporary mining settlement (built in 2011) located 17 km
northwest from Bolshoe Toko in the upper course of the Elga river. This settlement
was accessible by off-road vehicles we used to reach the lake, partly along
temporary winter roads (frozen rivers and lakes) in March 2013. The exploitation of
the El'ginsky coal deposits, planned for a productivity of 15-20 million tons year$^{-1}$
(Konstantinov, 2000), will strongly affect the lake and its catchment. The territory of
the watershed will increasingly be damaged and contaminated by off road vehicles
and rain fall will produce muddy water which potentially can cause lake pollution
(Sobakina and Solomonov, 2013).
The lake basin is adjoined to the northern slope of the eastern Stanovoy
mountain range in a depression of tectonic and glacial origin between two
northwest-trending right-lateral strike-slip faults (Imaeva et al., 2009). A southward
thrust fault runs along the southern border of the lake separating the Precambrian
igneous rocks in the south from sandstones and mudstones of the Mesozoic
Tokinski Plateau in the north. The Stanovoy mountain range in the southern




catchment of the lake consists mainly of highly mafic granulites and other high-pressure metamorphic rock types (Rundqvist and Mitrofanov, 1993). At its north-eastern margins the lake is bordered by moraines of three different glacial sub-periods (Kornilov, 1962) (Fig. 2).

The study area is situated within the East Siberian continental temperate climate zone exhibiting taiga vegetation (boreal forests) and fragments of steppes and a predominant westerly wind system (Shahgedanova, 2002). The meteorological station in Yakutsk has recorded historical climate data (Gavrilova, 1993). In the 19th Century the mean annual temperature was circa -11° to -11.5°C. During the 20th Century temperatures have increased to around -10.2°C, in parallel with an increase in precipitation from 205 to 250 mm per year. The meteorological station "Toko" located approximately 10 km northeast of the lake, however, recorded mean annual air temperatures of -11.2°C (January min. -65°C, July max. +34°C, annual precipitation 276-579 mm). Measurements taken directly at the lake were lower, indicating the influence of cold water from the Stanovoy mountain range in summer and the high volume of ice during wintertime (Konstantinov, 2000). Since the average air temperature in southern Yakutia increases with height (temperature inversion of ~2°C 100 m$^{-1}$), permafrost can be locally discontinuous where taliks (unfrozen zones) underneath topographically high and deep lakes penetrate the permafrost zone (Konstantinov, 1986). As observed in 1971 (Konstantinov, 2000) ice cover lasts at least partly until mid-July.





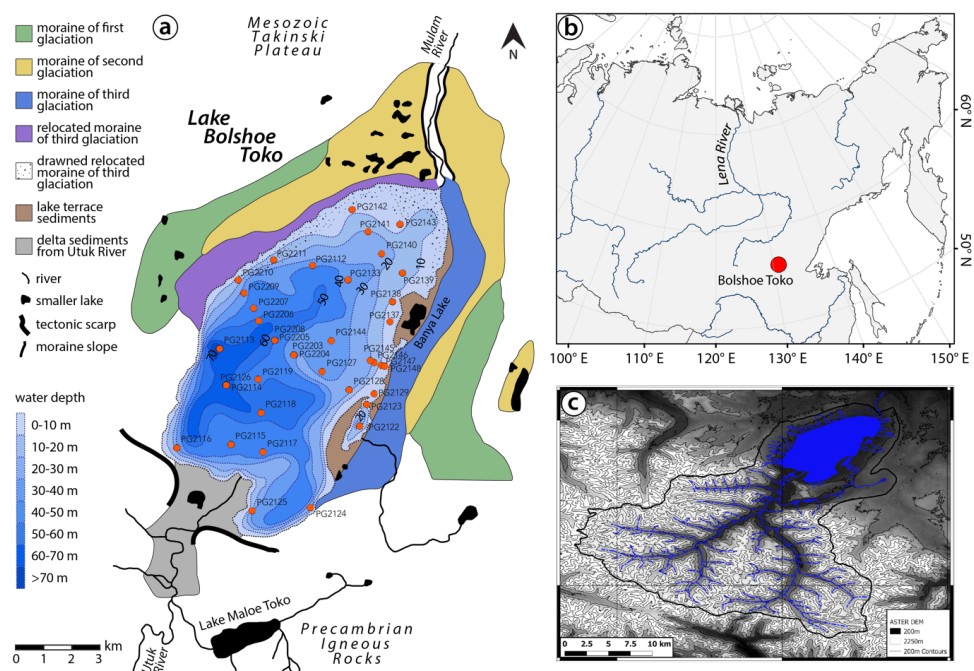

**Fig. 1** Lake Bolshoe Toko study site. **a** Geological map, bathymetry and moraines. Map compiled using data
from Konstantinov (2000) and Kornilov (1962). **b** Overview map of Siberia. World Borders data are derived from
http://thematicmapping.org/downloads/world_borders.php and licensed under CC BY-SA 3.0. **c** Catchment area
around Bolshoe Toko delineated from the ASTER GDEM V2 model between the latitudes N54° and N56° and
longitudes E130° to E131° (1) (Meyer et al., 2011) and a corresponding multispectral Landsat 8 OLI TIRS
satellite image using QGIS (QGIS-Team, 2016). Most of the river catchment is located in the igneous
Precambrian Stanovoy mountain range supplying the southern part of the lake with water and sediment. The
shallower northeastern part of the lake is influenced by the surrounding moraines and Mesozoic sand- and
mudstones.

## 3   Materials and methods
### 3.1   Field work
Field work was conducted during the German-Russian expedition "Yakutia 2013"
between March 19th to April 14th 2013 by the Alfred Wegener Institute Helmholtz
Centre for Polar and Marine Research (AWI) and the North Eastern Federal State
University in Yakutsk (NEFU). Lake basin bathymetry was measured with a portable
Echo Sounder. Water samples for hydrochemical analyses of the water column and
the ice layer were collected prior to sediment coring using a UWITEC water
sampler. Water samples were analysed in situ using a WTW Multilab 340i for pH,
conductivity, and oxygen values at the day of retrieval during field work. A sub-
sample of the original water was passed through a 0.45 $\mu$m cellulose-acetate filter,
stored and transported in 60-ml Nalgene polyethylene bottles for subsequent anion



and cation analyses in AWI laboratories in autumn 2013. Cation samples were
acidified during field work with $HNO_3$, suprapure (65%) to prevent microbial
conversion processes and adsorptive accretion.
At 42 sites within the lake, short cores containing intact sediment surface material
were retrieved using an UWITEC gravity corer. Water depth at sampling sites was
measured using either a hand-held HONDEX PS-7 LCD digital sounder and/or the
cord of the coring device when the lake ice cover disturbed the signal. The sediment
was identified as clayish silt deposits with predominant dark (black) color and a
weak smell of hydrogen sulphide, a sticky and viscous mud mixed with plant and
other organic residues. The uppermost ca. 2 cm at some sites had a dark red
colouring indicating the redox boundary between oxygenated and anoxic sediments.
We identified the uppermost 0.5 cm of short cores as surface sediments and
subsampled these layers onsite during fieldwork to avoid sediment mixture during
transport. Sediment samples were transported in sterile "Whirl-Pak" bags and
sediment cores were transported in plastic liners to the AWI laboratories in
Potsdam, Germany, and stored at 4°C in a dark room for further analyses and as
back-up.

## 3.2 Laboratory analyses

### 3.2.1 Hydrochemistry

From the water samples anions were analysed using ion chromatography
(Dionex DX 320) and cations were determined using inductively coupled plasma–
optical emission spectrometry (ICP-OES, Perkin-Elmer Optima 8300DV Perkin-
Elmer – Optical Emission Spectrometer. Hydrogen carbonate concentrations were
measured by titration with 0.01 M HCl using an automatic titrator (Metrohm 794
Basic Titrino).
Stable hydrogen and oxygen isotope analyses were carried out with Finnigan
MAT Delta-S mass spectrometers with two equilibration units using common
equilibration techniques (Meyer et al., 2000), and given as $\delta^{18}O$ and $\delta D$ in ‰ vs.
VSMOW (Vienna Standard Mean Ocean Water). The d excess ($d = \delta D - 8\delta^{18}O$) is
indicative for evaporation conditions in the moisture source region (Dansgaard,
1964;Merlivat and Jouzel, 1979).

### 3.2.2 X-ray fluorescence and X-ray diffractometry

The elemental composition of 20 freeze-dried and milled surface samples was semi-
quantitatively analysed by X-ray fluorescence (XRF) using a novel single sample
modification for the AVAATECH XRF core scanner at AWI Bremerhaven. A
Rhodium X-ray tube was warmed up to 1.75mA and 3 mA with a detector count time
of 10s and 15s for elemental analysis at 10kV (No filter) and 30kV (Pd-Thin filter)



respectively. The average modelled chi square values ($\chi^2$) of measured peak
intensity curve fitting for the relevant elements were variable, but generally low (Zr =
0.92, Mn = 1.49, Fe = 2.32, Ti = 1.53, Br = 3.65, Sr = 4.79, Rb = 4.98, Si = 16.11).
Values above 3 were ascribed to suspiciously high count rates from sample PG2133
which was subsequently excluded from XRF interpretation. The relatively low
amount of total sample material available did not facilitate the removal of organic
matter before prior to sample measurement and may have contributed to the
variable modelled chi square values.
As interpretation of raw device obtained element intensities (in counts per second,
cps) is problematic due to non-linear matrix effects and variations in sample density,
water content and grain-size (Tjallingii et al., 2007), cps values were transformed
using a centred-log ratio transformation (CLR). Element ratios were calculated from
raw cps values and transformed using an additive-log ratio transformation (ALR)
(Weltje and Tjallingii, 2008).
The mineralogical composition of 32 freeze-dried and milled samples was
analysed by standard X-ray diffractometry (XRD) using a Philips PW1820
goniometer at AWI Bremerhaven applying Cobalt-Potassium alpha (CoKα) radiation
(40 kV, 40 mA) as outlined in Petschick et al. (1996). The intensity of diffracted
radiation was calculated as counts of peak areas using XRD processing software
MacDiff 4.0.7 (freeware developed by R. Petschick in 1999). Individual mineral
contents were expressed as percentages of bulk sediment XRD counts (Voigt,
2009). Mineral inspection focused on quartz, plagioclase and K-feldspar,
hornblende, mica, and pyrite. Clay minerals involved kaolinite, smectite and chlorite.
Accuracy of the semi-quantitative XRD method is estimated to be between 5 and
10% (Gingele et al., 2001).

**3.2.3  Grain-size, carbon and nitrogen analyses**
In order to gain high-resolution information on the grain-size distribution, organic
material was removed from 32 surface sediment samples by hydrogen peroxide
oxidation over four weeks on a platform shaker. Two homogenised subsamples
were weighted and 93 subclasses between 0.375 and 2000 µm were measured
using a Coulter LS 200 Laser Diffraction Particle Analyser. Grain-size fractions
coarser than 2 mm were sieved out, weighted and added to the volume percentage
data afterwards to indicate the proportion of gravel.
Total carbon (TC) and total nitrogen (TN) of 35 freeze-dried and milled samples
was quantified by heating the material in small tin capsules using a Vario EL III CNS
analyser and total organic carbon (TOC) was measured using a Vario MAX C in per
cent by weight (wt%). The measurement accuracy was 0.1 wt% for TOC and TN,
and 0.05 wt% for TC. TOC and TN were compared to calculate the $TOC/TN_{atomic}$





ratio by multiplying with the ratio of atomic weights of nitrogen and carbon following
Meyers and Teranes (2002).
The stable carbon isotope composition $\delta^{13}$C of the total organic carbon fraction
was measured in 15 samples using a Finnigan Delta-S mass spectrometer. Dried,
milled and carbonate-free (HCl treated) samples were combusted in tin capsules to
$CO_2$. Results are expressed as $\delta^{13}$C values relative to the PDB standard in parts per
thousand (‰) with an error of ±0.15%.

### 3.2.4  Diatoms

23 samples were prepared for diatom analysis following the standard procedure
described by Battarbee et al. (2001). To calculate the diatom valve concentration
(DVC) $5\times10^6$ microspheres were added to each sample following organic removal
with hydrogen peroxide. Diatom slides were prepared on a hot plate using Naphrax
mounting medium. For the identification of diatoms to the lowest possible taxonomic
level we used several diatom flora including Lange-Bertalot et al. (2011), Lange-
Bertalot and Metzeltin (1996), Krammer and Lange-Bertalot (1986-1991) and
Lange-Bertalot and Genkal (1999). For rare taxa (i.e. *Pliocaenicus*) literature
research was applied in scientific papers, including Cremer and Van de Vijver
(2006) and Genkal et al. (2018). A minimum of 300 (and up to 400) diatom valves
were counted in each sample using a Zeiss AXIO Scope.A1 light microscope with a
Plan-Apochromat 100×/1.4 Oil Ph3 objective at 1000x magnification. Identification of
small diatom species was verified using a scanning electron microscope (SEM) at
the GeoForschungsZentrum Potsdam.
During counting of diatom valves, chrysophycean stomatocysts and *Mallomonas*
were counted but not further taxonomically identified. Count numbers were used to
estimate the chrysophyte cyst to diatom index (C:D) and *Mallomonas* to diatom
index (M:D) relative to counted diatom cells (Smol, 1984;Smol and Boucherle,
350  1985).

### 3.2.5  Oxygen isotopes of diatom silica

To analyze the oxygen isotope composition from diatom silica ($\delta^{18}$O$_{diatom}$) from 9
representative surface samples, a purification procedure including wet chemistry (to
remove organic matter and carbonates) and heavy liquid separation was applied for
the fraction <10 $\mu$m following the method described in Chapligin et al. (2012). After
freeze-drying the samples were treated with $H_2O_2$ (32%) and HCl (10%) to remove
organic matter and carbonates and wet sieved into <10 $\mu$m and >10 $\mu$m fractions.
Four multiple heavy liquid separation (HLS) steps with varying densities (from 2.25
to 2.15 g/cm3) were then applied using a sodium polytungstate (SPT) solution



before being exposed to a mixture of $HClO_4$ (65%) and $HNO_3$ (65%) for removing
any remaining micro-organics.
To remove exchangeable hydrous groups from the diatom valve structure
(amorphous silica $SiO_2$ * $nH_2O$), inert Gas Flow Dehydration was performed
(Chapligin et al., 2010). Oxygen isotope analyses were performed on dehydrated
samples using laser fluorination technique (with $BrF_5$ as reagent to liberate $O_2$) and
then directly measured against an oxygen reference of known isotopic composition
using a PDZ Europa 2020 mass spectrometer (MS2020, now supplied by Sercon
Ltd., UK). The long-term analytical reproducibility (1σ) is ±0.25 ‰ (Chapligin et al.,
370 2010).
Every fifth sample was a biogenic working standard to verify the quality of the
analyses. For this, the biogenic working standard BFC calibrated within an inter-
laboratory comparison was used (Chapligin, 2011). With a $\delta^{18}O$ value of +29.0±0.3
‰ (1σ) BFC (this study: +28.7±0.17 ‰, n=49) is the closest diatom working
standard to the Bolshoe Toko samples ($\delta^{18}O$ values range between +22 and +24 ‰)
available. A contamination correction was applied to $\delta^{18}O_{diatom}$ using a geochemical
mass-balance approach (Chapligin et al., 2012;Swann et al., 2007) determining the
contamination end-member by analysing the heavy fractions after the first heavy
liquid separation resulting in $Al_2O_3$=16.2±1.3 % (via EDX; n=9) and $\delta^{18}O$=8.5±0.8 ‰
(n=6).

### 381 3.2.6 Chironomids

Treatment of 18 sediment samples for chironomid analysis followed standard
techniques described in Brooks et al. (2007). Subsamples of wet sediments were
deflocculated in 10 % KOH, heated to 70 °C for up to 10 minutes, to which boiling
water was added and left to stand for up to another 20 minutes. The sediment was
passed through stacked 225 and 90 $\mu$m sieves. Chironomid larval head capsules
were picked out of a grooved Bogorov sorting tray under a stereomicroscope at 25-
40x magnifications and were mounted in Hydromatrix two at a time, ventral side up,
under a 6 mm diameter cover slip. From 48 to 117 chironomid larval head capsules
were extracted from each sample, to capture the maximum diversity of the
chironomid population. Chironomids were identified to the highest taxonomic
resolution possible with reference to Wiederholm (1983) and Brooks et al. (2007).
Information on the ecology of chironomid taxa and groups was taken from Brooks et
al. (2007), Pillot (2009) and Nazarova et al., (2011;2015;2008;2017b)). Ecological
information of the taxa associated to biotopes (littoral, profundal), water velocity
(standing, running water), and relation to presence of macrophytes were taken from
Brooks et al. (2007) and Pillot (2009). T July optima of chironomids were taken from
Far East (FE) chironomid-based temperature inference model (Nazarova et al.,
2015). The Far East (FE) chironomid-based temperature inference model (WA-PLS,



2 components; $r^2$ boot = 0.81; RMSEP boot = 1.43 °C) was established from a modern calibration data set of 88 lakes and 135 taxa from the Russian Far East (53–75°N, 141–163°E, T July range 1.8 – 13.3 °C). Mean July air temperature for the lakes from the calibration data set was derived from (New et al., 2002). All modern and chironomid-inferred temperatures were corrected to 0 m.a.s.l. using a modern July air temperature lapse rate of 6 °C km$^{-1}$ (Livingstone et al., 1999;Heiri et al., 2014).

### 3.3  Statistical analyses

Species richness and the Simpson diversity on diatom and chironomid data were estimated after sample-size normalization using a rarefaction analysis in the iNext package in R. Diatom valve preservation was measured and calculated as the f-index (Ryves et al., 2001). Diatom valve concentration was estimated as the number of valves per gram dry sediment following Battarbee and Kneen (1982).

Detrended Correspondence Analysis (DCA) with detrending by segments was performed on the chironomid and diatom data (rare taxa downweighted) to determine the lengths of the sampled environmental gradients, from which we decided whether unimodal or linear statistical techniques would be the most appropriate for the data analysis (Birks, 1995). For diatom data the gradient lengths of the species scores were 2.07 and 1.49 standard deviation units (SDU) for DCA 1 and 2, respectively, suggesting that lineal numerical methods should be used. A Principal Component Analysis (PCA) was used to explore the main taxonomic variation of the data (ter Braak and Prentice, 1988). The gradient lengths of chironomid species scores were 3.78 and 4.12 SDU indicating that numerical methods based on a unimodal response model should be more appropriate to assess the variation structure of the chironomid assemblages (ter Braak, 1995). However, test PCA performed on chironomid data showed that lineal method captures more variance of species data (ESM, Table a) therefore we further applied lineal methods for both, chironomid and diatom data. In order to summarize the response of lacustrine biota to abiotic, physicochemical explanatory variables, a redundancy analysis (RDA) was performed on diatom and chironomid data in comparison to environmental variables (Fig. 2 and 3).

Initially, all environmental variables shown in the Table 1 were used in RDA to assess the relationships between the distribution of bioindicator taxa and abiotic habitat parameters. Additionally we include in the analysis the presence/absence of the submerged vegetation, distances of the sampling stations from the shore and from the inflowing rivers. Variance inflation factors (VIF) were used to identify intercorrelated variables. Environmental variables with a VIF greater than 20 were eliminated, beginning with the variable with the largest inflation factor, until all remaining variables had values < 20 (ter Braak and Smilauer, 2012). A set of RDAs



was performed on chironomid and diatom data with each environmental variable as
the sole constraining variable. The percentage of the variance explained by each
variable was calculated and statistical significance of each variable was tested by a
Monte Carlo permutation test with 999 unrestricted permutations. Significant
variables (P≤ 0.05) were retained for further analysis.
DCA, PCA and RDA were performed using CANOCO 5.04 (ter Braak and
Smilauer, 2012).
Percentage abundances of the chironomid taxa that are absent or rare in the
modern calibration data set were calculated at each sampling site in order to see the
distribution of the taxa that could potentially hamper a T July reconstruction in case
of palaeoclimatic study that could be done at each of the sampling sites. It is known
that less reliability should be placed on the samples in which more than 5% of the
taxa are not represented in the modern calibration data or more than 5% of the taxa
are rare in the modern calibration dataset (i.e. Hill's N2 less than 5) (Heiri and Lotter,
2001;Self et al., 2011).
To assess the relative contribution of different sedimentary processes to the bulk
sediment, such as fluvial or aeolian transport (Wang et al., 2015;Biskaborn et al.,
2013b) a statistical end-member analysis on grain-size data was performed using
the MATLAB modelling algorithm of Dietze et al. (2012). In this method, individual
grain-size populations identified as end-member loadings (vol%, Fig. 4) as well as
their contributions to the bulk composition identified as scores (%) were derived by
eigenspace analysis, weight transformation, Varimax rotations and different scaling
procedures.
A Pearson correlation matrix of the main important variables (Fig. 5) was
calculated using the basic R core (R Core Team, 2012) and plotted using *corrplot*. A
p-value adjustment was applied to only assign colours to values that revealed p
<0.05. To identify the pattern, the correlation matrix was reordered according to the
correlation coefficient. Exceptional sites within the heterogenic lake system lead to
disturbance of good correlation coefficients within areas along natural borders, e.g.
water depth isobaths.
To guarantee the sustained availability of our research (Elger et al., 2016), the
data will be uploaded and freely accessible in the PANGAEA repository.





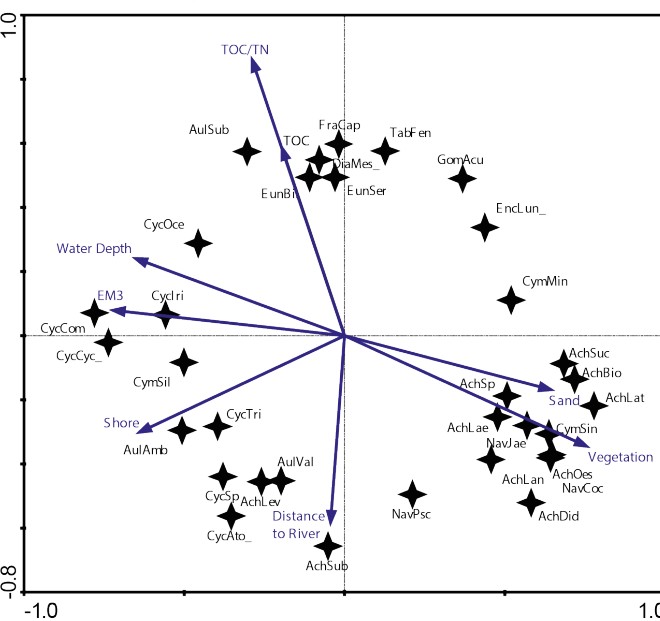

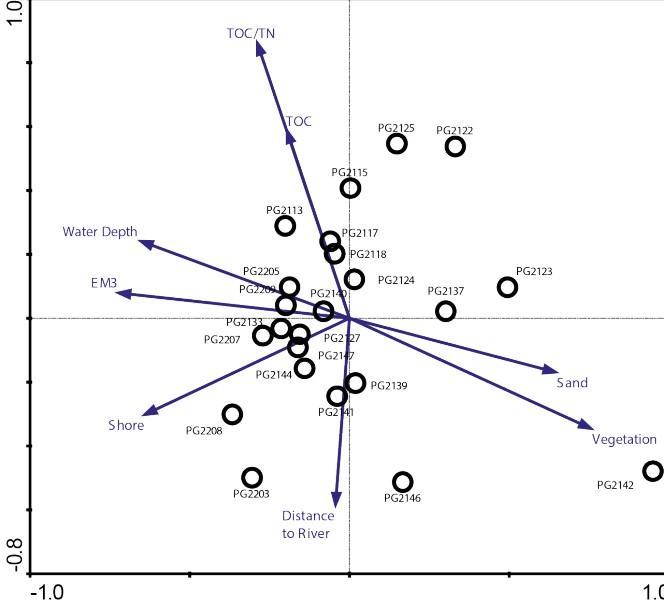

**Fig. 2** RDA biplots of diatoms in the surface sediments of Lake Bolshoe Toko. (a) Common diatom taxa and
significant environmental variables. (b) Diatom sampling sites and significant environmental variables.




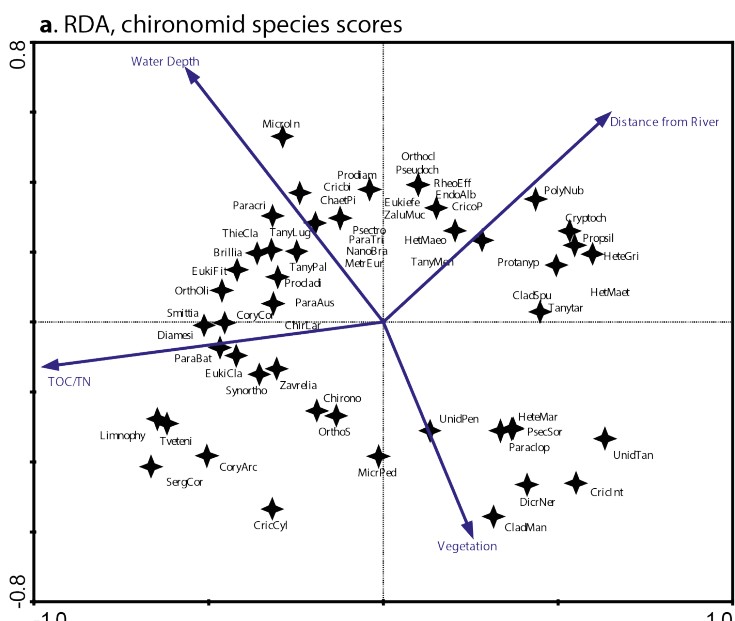

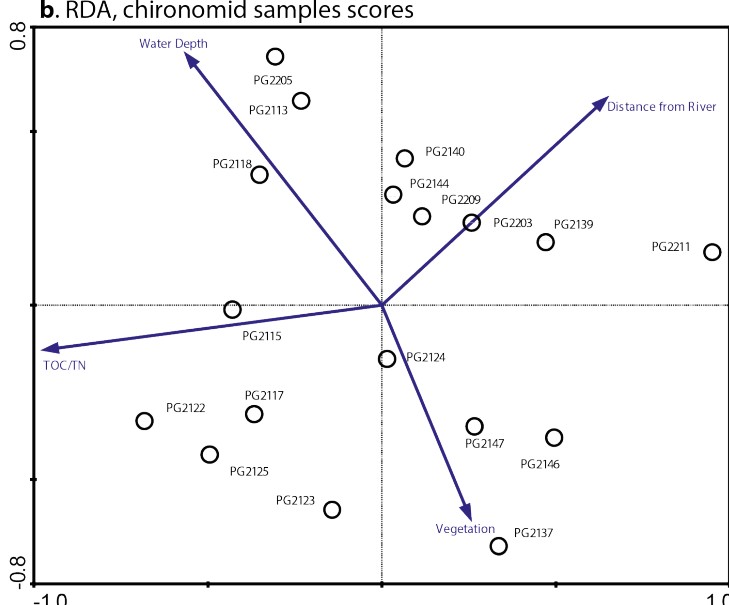

**Fig. 2** RDA biplots of chironomids in the surface sediments of Lake Bolshoe Toko. (a) Common chironomid taxa
and significant environmental variables. (b) Chironomid sampling sites and significant environmental variables.





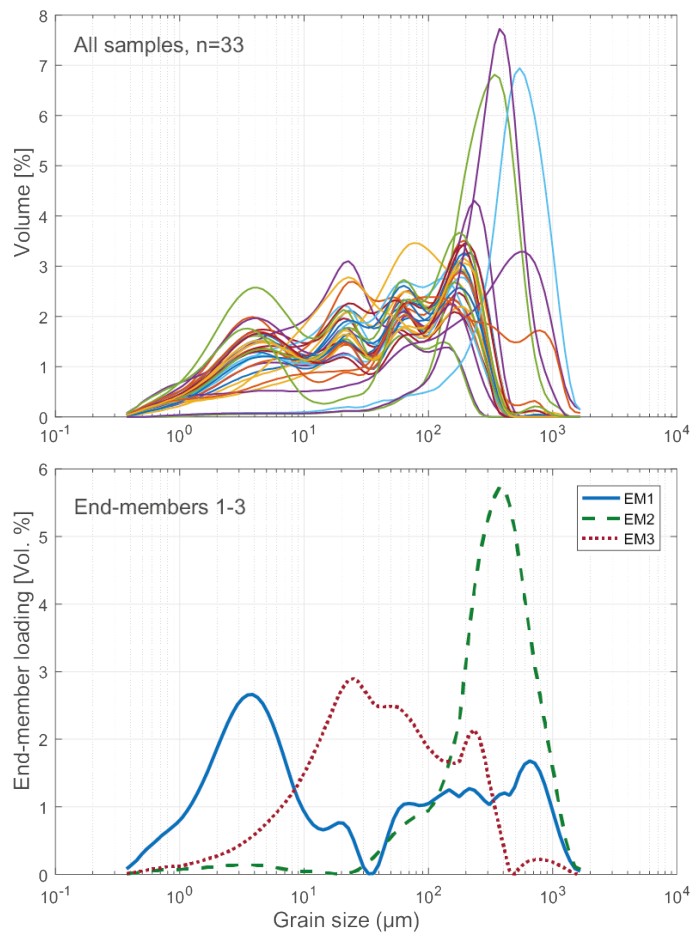

**Fig. 4** Endmember analysis grain-size distributions in 33 samples from Lake Bolshoe Toko.





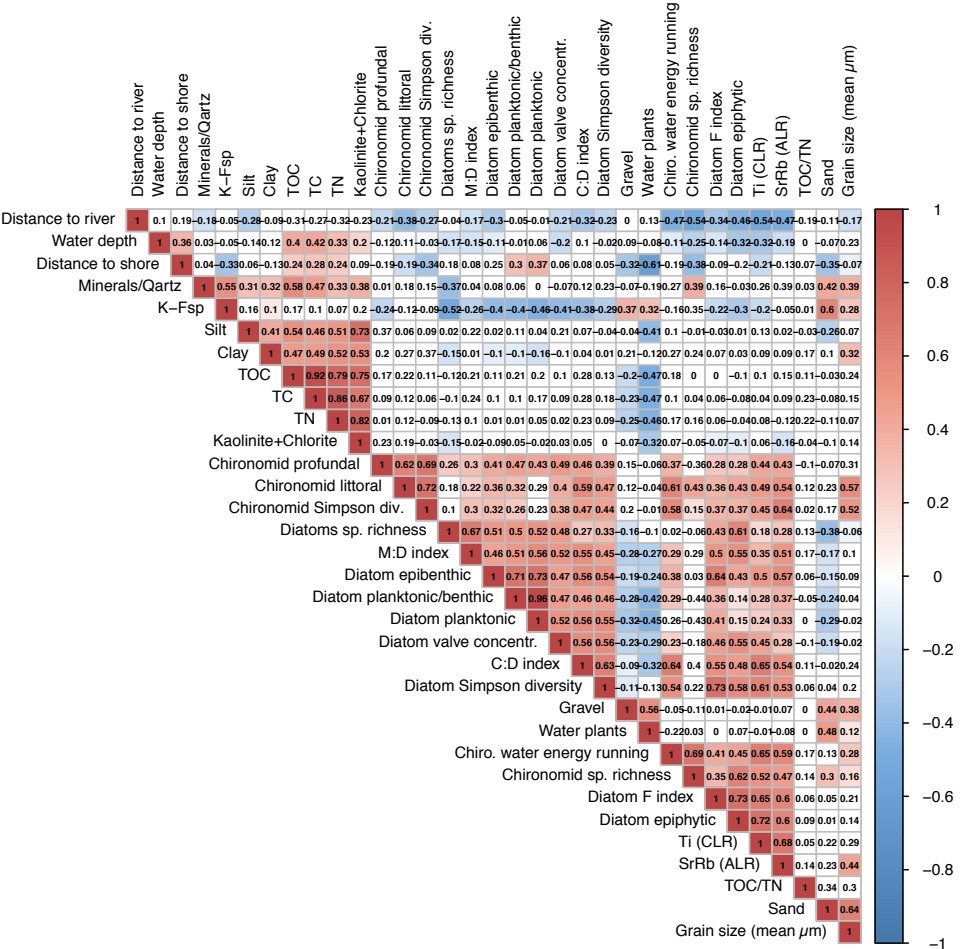

**Fig. 5** Pearson correlation matrix of selected environmental parameters. Positive correlations indicated in red, negative correlations indicated in blue. A p values adjustment was applied and only values of <0.05 used to assign colours.

## 4  Results

### 4.1  Water chemistry

The sampled surface waters of Bolshoe Toko (Table 1) are well saturated in $O_2$ (101-113 %) with a pH-value in the neutral range (6.8 – 7.2). The electrical conductivity is very low for all waters, though the lagoon shows a slightly higher conductivity (67.8 µS/cm) than the rest of the samples (35.1 – 39.1 µS/cm). Traces of Al (mean 72 µg/L), Fe (mean 46.6 µg/L), and Sr (mean 37.1 µg/L) were found. However, there was no evidence for significant concentrations of environmental



relevant elements (Pb, Cr, V, Co, Ni, Cu). The concentrations of sulfate ($SO_4^{2-}$) was
2.35 mg/l on average but lower in the lagoon (0.51 mg/l). The concentrations of
nitrate ($NO_3^-$) was 0.76 mg/l, but lower in the lagoon (0.29 mg/l). $HCO_3^-$ was 37.52
mg/l in the lagoon and 14.9 mg/l on average in the rest of the samples. There was
no phosphorus found in any of the samples. Overall the water can be characterized
as water of the Ca-Mg-HCO3 type.
The mean stable isotope composition of Bolshoe Toko lake surface waters at the
six coring positions is -18.7‰ for $\delta^{18}O$, -140.2‰ for $\delta D$ and 9.5‰ for d-excess,
respectively. A relatively uniform isotopic composition of $\delta^{18}O$ = -18.58±0.15‰ ($\delta D$ =
-139±0.7‰) was observed for the main Bolshoe Toko waters, whereas the lagoon
(PG2122) displays slightly more negative $\delta^{18}O$ ($\delta D$) values of -19.2‰ (-145‰).
Water depth profiles were taken during the March 2013 expedition from the deepest
part of the lake (PG2108, water depth 70m) and in the lagoon (PG2122, 18m) as
well as in August 2012 (sample site near the western shoreline, 37m). The
temperature was determined in the field and the samples analysed for isotopes
($\delta^{18}O$, $\delta D$, see Fig. 6). The mean isotopic composition of the water profile at PG2208
stabilizes from 10 m downward ($\delta^{18}O$ = -18.2 ± 0.2 ‰) and is slightly heavier than
the surface samples ($\delta^{18}O$ = -18.6) due to isotopic fractionation during ice formation.
In contrast, the lagoon shows a lighter isotope composition ($\delta^{18}O$ = -18.9 ± 0.2 ‰)
than the main lake basin. Samples taken in August 2012 close to the western
shoreline show a similar mean value down the water column ($\delta^{18}O$ = -18.2 ±0.3 ‰)
but no change in the upper samples as seen in PG2208. A similar mean isotopic
composition indicates negligible evaporation effects and no strong seasonal change.
This is typical for through-flow lakes (Mayr et al., 2007). Generally, a higher variation
is observed in the August record. Meteorological data from the nearby weather
station (Toko RS, 10 km northward) recorded heavy rainfall for August 2012 (25 mm
above the long term mean of 83 mm). Such precipitation events could cause
temporary isotopic stratification or a variation in the isotopic signal throughout the
water column. Due to ongoing mixing, these variations are then evened. In
conclusion, variations in the isotopic composition throughout the August profile are
more a temporal phenomenon and not characteristic for Bolshoe Toko. All samples
are positioned close to the global mean water level (GMWL, Fig. 6) indicating an
unaltered precipitation signal without significant evaporation.




**Table 1**: Hydrochemical parameters from surface water samples of Lake Bolshoe Toko.

| | Unit | PG2207 | PG2208 | PG2122* | PG2124 | PG2125 | PG2126 | Average |
|---|---|---|---|---|---|---|---|---|
| pH | | 6.95 | 6.81 | 6.99 | 7.05 | 7.24 | 7.13 | **7.03** |
| Conductivity | µS/cm | 38.40 | 35.10 | 67.80 | 37.90 | 39.10 | 36.60 | **42.48** |
| Oxygen | % | 100.9 | 110.9 | 108.4 | 110.4 | 108.9 | 113.1 | **108.8** |
| Al | µg/L | 82.18 | 79.88 | 43.19 | 75.54 | 73.31 | 77.62 | **71.95** |
| Ba | µg/L | < 20 | < 20 | < 20 | < 20 | < 20 | < 20 | **< 20** |
| Ca | mg/L | 5.00 | 4.73 | 9.01 | 4.68 | 4.91 | 4.70 | **5.51** |
| Fe | µg/L | 24.56 | 26.90 | 147.56 | 24.96 | 33.13 | 22.22 | **46.55** |
| K | mg/L | 0.37 | 0.36 | 0.40 | 0.36 | 0.40 | 0.37 | **0.38** |
| Mg | mg/L | 1.16 | 1.09 | 2.71 | 1.07 | 1.13 | 1.10 | **1.38** |
| Mn | µg/L | < 20 | < 20 | < 20 | < 20 | < 20 | < 20 | **< 20** |
| Na | mg/L | 0.74 | 0.78 | 1.61 | 0.77 | 0.79 | 0.76 | **0.91** |
| P | mg/L | < 0,10 | < 0,10 | < 0,10 | < 0,10 | < 0,10 | < 0,10 | **< 0,10** |
| Si | mg/L | 2.11 | 1.93 | 3.01 | 1.98 | 2.05 | 2.05 | **2.19** |
| Sr | µg/L | 27.40 | 25.86 | 90.57 | 26.28 | 26.29 | 26.07 | **37.08** |
| Pb | µg/L | < 25 | < 25 | < 25 | < 25 | < 25 | < 25 | **< 25** |
| Cr | µg/L | < 20 | < 20 | < 20 | < 20 | < 20 | < 20 | **< 20** |
| V | µg/L | < 20 | < 20 | < 20 | < 20 | < 20 | < 20 | **< 20** |
| Co | µg/L | < 20 | < 20 | < 20 | < 20 | < 20 | < 20 | **< 20** |
| Ni | µg/L | < 20 | < 20 | < 20 | < 20 | < 20 | < 20 | **< 20** |
| Cu | µg/L | < 20 | < 20 | < 20 | < 20 | < 20 | < 20 | **< 20** |
| Zn | µg/L | < 20 | < 20 | < 20 | < 20 | < 20 | < 20 | **< 20** |
| Fluoride | mg/l | < 0,05 | < 0,05 | < 0,05 | < 0,05 | < 0,05 | < 0,05 | **< 0,05** |
| Chloride | mg/l | 0.51 | 0.51 | 0.55 | 0.60 | 0.60 | 0.58 | **0.56** |
| Sulfate | mg/l | 2.72 | 2.47 | 0.51 | 2.66 | 2.95 | 2.77 | **2.35** |
| Bromide | mg/l | < 0,10 | < 0,10 | < 0,10 | < 0,10 | < 0,10 | < 0,10 | **< 0,10** |
| Nitrate | mg/l | 0.82 | 0.82 | 0.29 | 0.82 | 0.88 | 0.89 | **0.76** |
| Phosphate | mg/l | < 0,10 | < 0,10 | < 0,10 | < 0,10 | < 0,10 | < 0,10 | **< 0,10** |
| HCO3- | mg/l | 15.71 | 13.58 | 37.52 | 14.80 | 15.41 | 15.10 | **18.69** |
| δ18O | ‰ VSMOW | -18.72 | -18.5 | -19.21 | -18.63 | -18.43 | -18.71 | **-18.70** |
| δD | ‰ VSMOW | -140.2 | -138.5 | -145.3 | -139.4 | -138.2 | -139.3 | **-140.15** |
| d-excess | ‰ VSMOW | 9.6 | 9.4 | 8.4 | 9.7 | 9.2 | 10.4 | **9.45** |






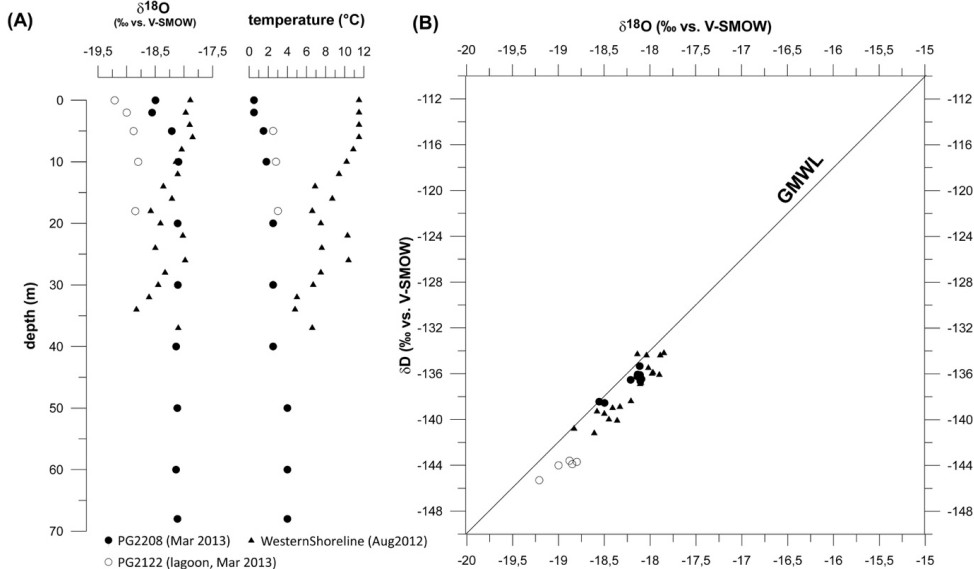

**Fig. 6**: (A) Profiles of water isotopes ($\delta^{18}O$) and temperature from different locations taken in August and March.
(B) $\delta^{18}O$-$\delta D$ diagram for various water samples. GMWL is the Global Meteoric Water Line (black line),

## 4.2 Physicochemical sediment composition

The typical surficial lake bottom sediments consist of either brown organic-enriched gyttja or sandy, organic-poor siliciclastic material. Sand contents ranged between 10.2 % and 96.2 % (mean 45.9 %, Fig. 7); silt contents ranged from 3.6 % to 83.3 % (mean 47.1 %); clay contents ranged from 0.2 % to 11.3 % (mean 5.8 %). Gravel was found only in four samples at the north eastern near-shore areas with contents of up to 13.1 %. The mean grain size ranged from 12 to 479 $\mu$m (mean 72 $\mu$m). The mean grain size generally correlated negatively with water depth (r -0.45). Mineral grains are composed mainly of quartz (32.7-76.2 %, mean 55.4 %), plagioclase (13.4-39.5 %, mean 26.2 %), K-feldspar (0.0-9.8 %, mean 5.6 %), and, to a smaller degree of pyrite (0.2-5.5 %, mean 3.3 %), hornblende (0.5-10.8 %, mean 3.1 %), mica (0.3-2.4 %, mean 1.1 %), and the clay minerals smectite, kaolinite and chlorite (together 0.0-4.6 %, mean 2.0 %). The spatial distribution of minerals (Fig. 7) revealed a generally decreasing gradient of minerals relative to quartz starting from the Utuk river delta (proximal) towards the northern areas (distal).

The CLR transformed XRF data (Fig. 8) revealed high proportions of Zr and intermediate to high Ti near the Utuk river inflow and at the northern and eastern shore proximal areas. Zr values are seen to decrease with increasing water depth





towards the lake centre with the exception of the shallow lagoon where low values are observed. Mn values are highest in the lake centre and at the very deep site at the western steep subaquatic slope, and intermediate at shallow areas close to the shore. A minimum in Mn is seen in the lagoon. Fe tends to be highest in the southern part of the lake basin, in the very shallow site in the north, and in the lagoon. Br demonstrates an unclear distribution; however high values are found at 2 sites within the eastern lagoon that correspond to high TOC contents.

Additive log ratios (ALR) of Mn/Fe were variable with intermediate values found at sites surrounding the Utuk river inflow and low values within the lagoon and at basin central sites. High values were located at the deepest lake site as well as in the shallow north eastern region. Both Sr/Rb and Zr/Rb ratios demonstrated significantly high values directly in front of the Utuk river inflow that diminished with distance towards the basin center. Both Sr/Rb and Zr/Rb possessed intermediate to high values in the north eastern lake region and suppressed values within the lagoon. Si/Ti ratio values showed a trend from low in the southern lake region and lagoon to high in the northern lake region.

The contents of total organic carbon (TOC, Fig. 9) ranged from 0.1 % to 12.3 % (mean 4.9 %). Highest values appeared in the eastern area, intermediate values in the central parts, and lowest contents in the northern shallow water areas. The difference between TOC and total carbon is within the error of the devices and hence no inorganic carbon was detected. TOC contents and the TOC/TN ratios are highest near the Utuk river inflow in the southern part of the lake, in the lagoon, and in proximity to the eastern shoreline. $\delta^{13}C$ was exemplary measured in 15 samples and revealed maximum values at the eastern shore (-25.7 ‰) and minimum values everywhere else (-27.8 ‰).





**Fig. 7** Spatial distribution of the grain-size and mineral compositions of the surface sediments of Lake Bolshoe
Toko.



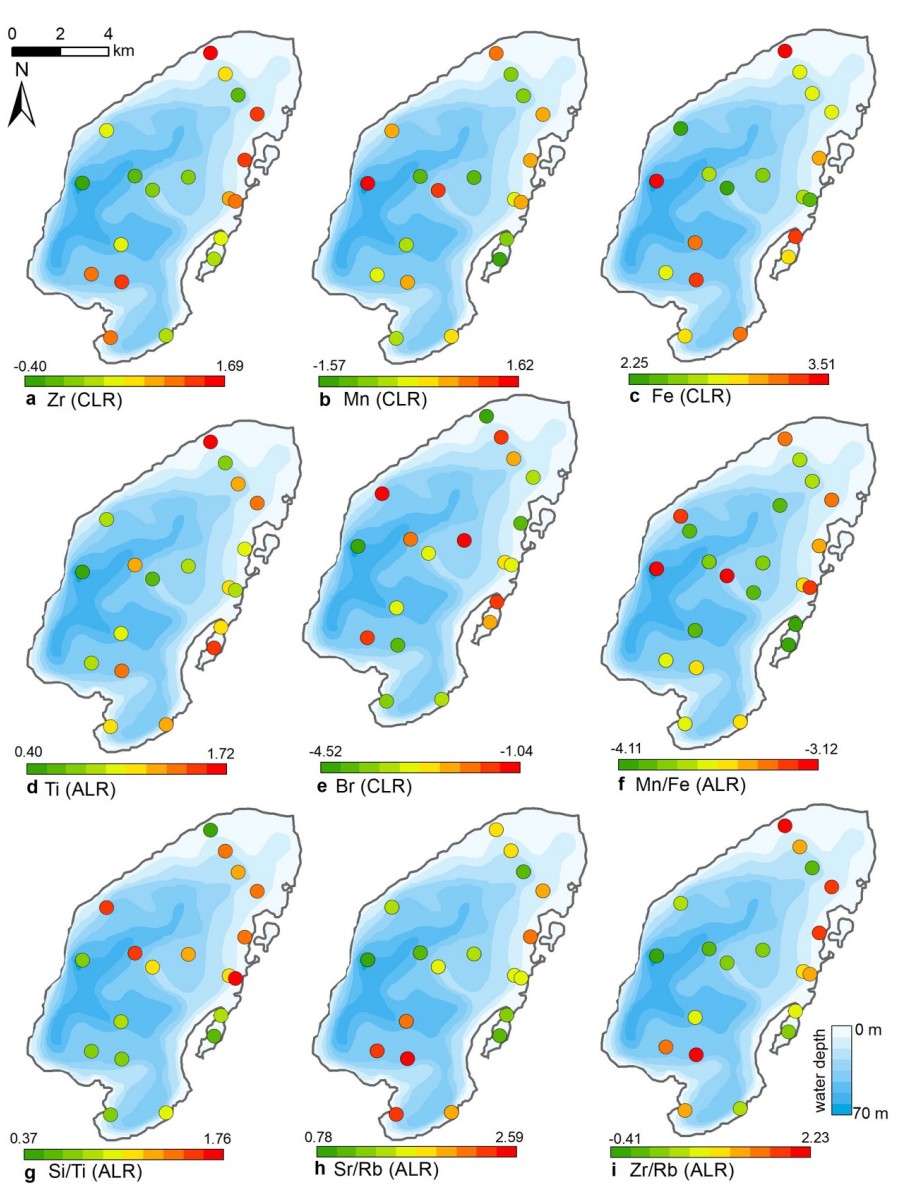

**Fig. 8** Spatial distribution of elements obtained from XRF measurements of surface sediments of Lake Bolshoe
Toko.



### 4.3 Diatoms

The diatom species assemblage in the analysed surface samples was generally represented by boreal and arcto-alpine types and varied distinctly within Lake Bolshoe Toko. In total, 142 diatom taxa were found in 23 sites, dominated by planktonic species *Pliocaenicus bolshetokoensis (Genkal et al., 2018)* (0.0-27.9 %, mean 14.7 %), *Cyclotella comensis* (0.0-23.1 %, mean 10.9 %), and benthic species *Achnanthidium minutissimum* (0.0-38.0 %, mean 11.8 %). The relative content of planktonic species (Fig. 9) was 2.0-73.7 % (mean 54.2 %), epiphytic species 19.2-83.9 % (mean 36.4 %), and epibenthic species 2.6-23.0 % (mean 9.3 %). The spatial distribution of the main taxa are shown in Fig. 10. Small benthic fragilarioid species are represented by 0.0-27.6 % (mean 7.4 %), Naviculoid species ranged from 3.3 % to 12.9 % (mean 7.2 %), and *Aulacoseira* species ranged from 0.0 % to 10.8 % (mean 4.5 %). *Pliocaenicus bolshetokoensis* maximal appearance was in the areas with deepest water depth in the southern part of the lake and in the eastern lagoon. *Cyclotella* species were more frequent in the central part of the lake and were not as strictly bound to water depth as *Pliocaenicus*. *Aulacoseira* species were distributed without clear patterns in the central part and less abundant in the northern shallow water areas. *Tabellaria* species were more frequent in shallow near-shore areas than in central and deep-water areas.

Achnanthoid (monoraphid) species were most abundant in near-shore areas, especially near the eastern lake terrace. Fragilarioid (araphid) species were common in the southernmost part near the inflow as well as in the lagoon. Other benthic species, i.e. *Navicula*, *Cymbella*, and *Eunotia* were generally more abundant in shallow near-shore areas than in deeper water areas.

In pelagic areas planktonic diatoms were generally more abundant principally in pelagic areas than epiphytic and epibenthic species. Epiphytic species, however, predominated in some shallow areas in the north and east parts of the lake. Epibenthic species occurred in smaller abundancies in shallow lake littorals. Together with an increased amount of non-planktonic species, the Simpson diatom species diversity was higher in northern and eastern parts of the lake. The chrysophytes index was higher near the river inflow in the south and along the river-like bathymetrical structure and in the lagoon, where another small river runs into the lake. The *Mallomonas* index was high near the inflow and in the central part, and it was low at near-shore areas in the north and east. Highest f-index, representing the best valve preservation, was found in the near shore areas, whereas lowest values were found at a shallow bathymetrical structure in the central part of the lake. The highest valve concentrations were observed in the central and northern lake basin.

The initial RDA with all environmental variables showed that the axes 1 and 2 explained 39.6 % of variance of diatom species data. After deleting all





intercorrelated variables, 13 parameters with VIFs <20 were left for manual
selection with Monte-Carlo test. It revealed 8 statistically significant (p≤0.05)
explanatory variables: TOC/TN, TOC, water depth, distance from River, distance
from the shore, presence of vegetation, Sand, and EM3, (ESM diatoms, Fig. 2).
Eigenvalues for RDA axes 1 and 2 constrained by eight significant environmental
variables constitute 81% and 59%, respectively, of the initial RDA, suggesting that
the selected significant variables explain the major variance in the diatoms data.
The RDA biplots of the species scores and sample scores (Fig. 2) show that diatom
species and sites are grouped according to the main environmental forcing
responsible for their spatial distribution. The clearest environmental signals in the
RDA are related to water depth, habitat preferences and river influence. The upper
left quarter of the biplot is strongly influenced by water depth, grain size (EM3), and
the ratio between TOC and TN. The species found next to water depth are
planktonic *Cyclotella* taxa, whereas *Aulacoseira* is closer to TOC/TN and the total
carbon content. In the lower right quarter epiphytic and benthic taxa prevail, i.e.
achnanthoid, naviculoid and cymbelloid taxa, associated to the presence of
vegetation and coarser (sand) substrate conditions. The distances to river and to
shore are crossing the lower left quarter and are associated to different planktonic
Cyclotella and achnanthoid taxa, while in the opposite direction, with increasing
Utuk river influence, fragilarioid taxa, *Eunotia, Tabellaria, and Gomophonema*
prevail, next to the high influence of TOC/TN.

The $\delta^{18}O_{diatom}$ averages +22.8 ‰ (min. +21.9 ‰, max. +2.6 ‰, n=9, Fig. 9) with a
spatial distribution of higher values around 23.3 ‰ in the deeper south-western part
of lake (PG2113, 2115, 2005) where as lowervalues of app. 22.3 ‰ prevail in the
shallower northern lake basin (PG2139, 2140, 2147, 2209). The two samples from
the lagoon show 22.2 ‰ in the shallower northern area and 23.6 ‰ in the deeper
part. Generally, the surface sediment $\delta^{18}O_{diatom}$ show a standard deviation of ± 0.6
‰ (1σ). Four samples from the southern part could not be purified well enough and
show contamination corrections >2 ‰.





**Fig. 9** Spatial distribution of organic properties and statistical parameters inferred from diatom assemblages in
the surface sediments of Lake Bolshoe Toko.



**Fig. 10** Spatial distribution of main diatom taxa in the surface sediments of Lake Bolshoe Toko.





## 4.4 Chironomids

In the surface sediment samples, we identified in total 79 chironomid taxa of
which 48 belonged to subfamily Orthocladiinae, 25 to subfamily Chironominae (15
from the triba Tanitarsini and 10 from the triba Chironomini), 4 taxa were from
subfamily Diamesinae and 2 from Tanypodinae.
The initial RDA with all environmental variables shows that the RDA axes 1 and 2
explained 46.7% of variance in taxon data. Most of the environmental parameters
appeared to be intercorrelated and after deleting one by one all redundant variables,
eight parameters with VIFs <20 remained for the further analysis. The manual
selection with Monte-Carlo test selection revealed 4 statistically significant (p≤0.05)
explanatory variables: TOC/N, water depth (WD), distance from River, and presence
of vegetation (Table 2). Eigenvalues for RDA axes 1 and 2 constrained by four
significant environmental variables were 0.200 and 0.150, respectively. They
constitute 70 and 85 % of the RDA performed on all environmental variables (0.289
and 0.177, respectively). This minor difference suggests that the four selected
variables sufficiently explain the major gradients in the chironomid data.
The RDA biplot of the sample scores shows that sites are grouped by their
location in relation to the major environmental variables (Fig. 11) and distribution of
chironomid taxa along the RDA axes reflects their ecological spectra. Table 2 and
Fig. 11 show median values of eco-taxonomical chironomid groups and their relation
to environmental parameters.
Sites most strongly influenced by the inflowing rivers are grouped in the lower
left quarter of the biplot, as the vector in the upper right quarter shows an increase
of the distance from the river mouth. In total 64 chironomid taxa have been found in
this group of sites, and of these 33 have been found only here. Chironomid fauna is
represented by mainly phytophilic littoral taxa from the Orthocladiinae genera
*Cricotopus, Orthocladius, Eukiefferiella,* and *Parakiefferiella* etc. (Fig. 11). Another
important feature of the fauna here is the presence of a relatively high amount of the
taxa characteristic for lotic environments, among which are several *Diamesa* taxa,
*Rheocricotopus effusus*-type, *Synorthocladius, Brillia*, and for lotic-lentic
environments like *Parakiefferiella bathophila*-type, P. *triguetra*-type, *Nanocladius
rectinervis*-type, *N. branchicolus*-type, several *Eukiefferiella* taxa, and
*Stictochironomus*.
The group in the opposite upper right quarter represents the northern part of
the lake situated far from the inflowing rivers. Here, mainly profundal taxa prevail,
i.e. *Procladius, Polypedilum nubeculosum*-type, *Cryptochironomus* (eurytopic*), and
*Heterotrissocladius maeaeri*-type 1 (acidophilic).

The lower right group of sites represent eastern shallow littoral with presence
of macrophytes. Species richness and proportion of semiterrestrial and littoral taxa
in this group is generally low. 29 chironomid taxa were found here and 5 taxa were





found only in this group of sites. Littoral taxa here are generally phytophilic:
*Cricotopus intersectus*-type, *C. cylindraceus*-type, *Dicrotendipes nervosus*-type
(mesotrophic), and *Cladotanytarsus mancus*-type and *Psectrocladius sordidellus*-
type (acid-tolerant mesotrophic). Most abundant profundal taxa here belong to the
acid-tolerant *Heterotrissocladius* genera represented by *H. macridus*-type, *H.*
*maeaeri*-type 1 and 2, *H. grimschawi*-type (acidophilic), and to the subfamily
Tanypodinae represented by *Procladius*. The sites grouped in the opposing upper
left quarter represent lotic and lotic-lentic taxa (*Diamesinae, Thenimaniella*
*clavicornis*-type, *Eukiefferiella claripennis*-type, *Eukiefferiella fittkaui*-type, several
*Orthocladius* taxa).


**Table 2**. Median representation of ecological chironomid groups in the modern FE chironomid based
training set (Nazarova et al., 2015) in relation to mean July temperature, biotopes, water velocity, and
presence of macrophytes (vegetation) in groups of sites reveled by the RDA. UN- unknown (all
specimens that were identified to subfamily level only due to bad heads preservation or no
information available); ST – semiterrestrial; L - littoral; P – profundal; R – river (lotic); S – standing
water (lentic); F – phytophilic; N – neutral.

| Group of sites | T optima, °C | | | | Biotope | | | | Water velocity | | | | Vegetation | | | Represen-tation in the modern FE training set |
|---|---|---|---|---|---|---|---|---|---|---|---|---|---|---|---|---|
| | 14-17 | 11-13 | 7-10 | UN | L-ST | L | P | UN | R | R-S | S | UN | F | N | UN | |
| River | 8.51 | 53.57 | 13.83 | 12.50 | 1.79 | 59.83 | 32.34 | 10.71 | 8.64 | 27.66 | 52.13 | 10.71 | 46.25 | 41.25 | 10.71 | 80.78 |
| Eu Littoral | 16.50 | 57.43 | 12.33 | 16.93 | 0.00 | 38.98 | 36.31 | 16.93 | 0.00 | 14.64 | 68.67 | 16.93 | 34.85 | 49.49 | 16.93 | 91.69 |
| Sub Littoral | 12.08 | 81.67 | 2.08 | 4.17 | 0.00 | 15.00 | 59.58 | 4.17 | 0.00 | 2.08 | 91.25 | 4.17 | 12.50 | 83.75 | 4.17 | 100 |
| Profundal | 0.00 | 72.73 | 0.00 | 22.73 | 0.00 | 9.09 | 50 | 18.18 | 0.00 | 9.09 | 72.73 | 18.18 | 9.09 | 72.73 | 18.18 | 78.03 |







**Fig. 11** Spatial distribution of chironomid taxa and inferred statistical parameters in the surface sediments of Lake Bolshoe Toko.



## 5 Discussion

### 5.1 Spatial control of abiotic and biogeochemical sediment components

Physical properties of particles within the surface sediments of Bolshoe Toko depend chiefly on transportation processes and the characteristics and availability of clastic compounds in the source areas in the lake catchment. The main catchment comprises the Stanovoy mountain range in the south channelled through the Utuk river into Bolshoe Toko. Accordingly, the lake experiences annual input of suspended material through a single source at the Utuk river mouth that likely is at its maximum during spring snow melt (Bouchard et al., 2013). The grain-size data and its end-members (Fig. 4 and 7) show that the proportions of sand, silt, and clay remain somewhat constant in proximity to the Utuk river inflow but change towards the north and at the lake shoreline. Whereas in the central northern lake basin the amount of silt increases, the proportions of sand increase along the northern shoreline on top of the drowned moraine. Gravel is only present in samples near the lake terraces in the east. The constant distribution in the south-central lake basin reflects the river input. Decreasing river influence and hence decreasing water transport energy with increasing distance from the river mouth leads to the observed predominance of finer grain-sizes (silt dominated) samples in the northern central parts of the lake. Sandy samples along the shoreline reflect direct input from the moraines around the northern part of the lake. Other relevant within-lake sedimentary processes include shore-erosion and inwash and winnowing of fine sediment grains by surface currents as well as alluvial processes and debris flows which continue basin ward as subaquatic flows. The restriction of gravel at the eastern shore can be attributed to the availability of source material and suitable transport pathways of coarser clasts from the third moraine. In consequence to the described lateral transport trajectories and local control factors within the lake, there is only weak negative correlation between mean grain size and water depth (r -0.45, Fig. 7 and 12).

The modelled end-member loadings of the observed grain-size classes (Fig. 4 and 7) indicate EM1 having the major peak in fine silt representing mainly fluvial sediment input. EM2 having its peak values in fine to medium sandy grain-size fractions and in the northern part of the lake points to depositional processes associated with the erosion of moraines distal from the river inflow, where the hydrological dynamics in the lake basin are weak. As shown by a weak positive correlation between EM3 and the concentration of diatom valves (r 0.44), EM3 likely represents both in-situ diatom valves, that could not be removed from the allochthonous sediment particles during our sample processing, and possibly ice-rafted debris that is transported over the lake after ice break up (Wang et al., 2015).





Intermediate concentration of TOC and high ratios of TOC/TN in the south as
compared to the north suggest differences in catchment characteristics, i.e. a
considerable allochthonous contribution of terrestrial plant material from the Utuk
river. This assumption is supported by previous findings that showed that non-
vascular plants, i.e. phytoplankton and other algae, usually have TOC/TN ratios
between ca. 5 and 10 while organic matter from vascular land plants has higher
values of about 20 (Meyers and Teranes, 2002). High values of TOC/TN in lake
sediment surfaces at river inflows have also been observed in other studies (Vogel
et al., 2010). $\delta^{13}C$ was generally low on average (-26.8) and only slightly higher at
the eastern shore (-25.7), suggesting a strong overall dominance of $C_3$ plants and
phytoplankton in the bulk organic matter fraction (Meyers, 2003). It is yet unclear to
what degree old and reworked organic carbon, e.g. from charcoal deposits, is
transported to the lake.
The distribution of elements from the XRF scanning data suggests strong abiotic
relationships to grain-size and mineral distributions. We focus on heavier elements
because lighter elements, even though they are commonly present in higher
concentrations, show potential contribution from multiple sources. Sr/Rb ratios and
Zr are negatively correlated with Kaolinite/Chlorite (r -0.73 and -0.85, respectively).
As described in Kalugin et al. (2007), Rb substitutes for K in clay minerals. The
Sr/Rb ratios do not show however a significant correlation with grain-size
parameters as found in other studies (Biskaborn et al., 2013b). We assume
therefore that Sr, as substituent for Ca, is influenced by multiple minerals
represented in different grain-size fractions, i.e. K-feldspar (r 0.45) and Hornblende
(r 0.24). Associated to high metamorphic grades in the Stanovoy mountains, Sr is
preferentially taken into the K-feldspar phase (Virgo, 1968). The Zr/Rb ratio on the
other hand, correlates well with the sand fraction (r 0.50) and with the mean grain
size (r 0.49), but negatively with silt (r -0.54) and clay (r -0.39). We explain this effect
by a higher diversity of minerals caused by the input of the Utuk river supplying the
lake basin with mafic Ca-rich metamorphic rocks from the Stanovoy mountains. The
strong influence of the Utuk river in the spatial distribution of physicochemical
sediment components is demonstrated by the decreasing gradient of minerals
relative to quartz starting from the Utuk river towards the northern lake basin (Fig.
7). The most representative indicator of grain size variations in surface sediments is
given by clr transformed values of Ti which correlate well with the sand fraction (r
0.74) and the mean grain size (r 0.88).
Si/Ti ratios have been used previously as a proxy for the biogenic silica content of
sediments (Melles et al., 2012). This stems from the fact that Ti is generally
attributed to detrital influx and Si to both detrital and biogenic (diatom) origins. At
Bolshoe Toko somewhat positive correlations between Si/Ti ratios, diatom valve
concentrations (r 0.36) and the ratio of planktonic to benthic diatoms (r 0.42)



suggests that Si/Ti may be useful to trace the relative portion of diatom valves in
intermediate grain-size fractions. Moreover, the Si/Ti ratio correlates significantly
with silt (r 0.81).
Mn/Fe ratios have often been ascribed to redox dynamics associated to bottom
water oxygenation processes (Naeher et al., 2013). In Bolshoe Toko, however, the
detrital input of ferrous minerals, i.e. pyrite, suggests that the Mn/Fe ratios cannot
directly be used effectively to track redox processes in the surface sediments. This
is supported by the correlation of Fe with the sand fraction (r 0.6) and grain-size (r
0.59). Accordingly, we found no significant correlations between Mn/Fe and other
abiotic or biotic proxies.
There is an uncertainty in the spatial distribution of elements measured by XRF
techniques. We attribute this lack of clear patterns to two main reasons: (1)
methodological hurdles to apply XRF techniques to surface sediments commonly
rich in water and organic material, and (2) multiple sources of the same elements
coming from minerogenic input, grain-size differences in individual samples and
different intensities of redox processes at different habitat settings. The high
variance of elements therefore should be seen as a representation of the high
complexity of this lake system rather than unequivocal validations or falsifications of
the applicability of XRF scanner data as an environmental proxy at Bolshoe Toko.

**5.2  Factors explaining the spatial diatom distribution**
Given the fact that diatoms react rapidly to environmental changes, different
driving factors influence the diatom distribution at different sites including
hydrochemical parameters, water depth, nutrients, and catchment vegetation type
(Pestryakova et al., 2018). Planktonic diatom species have appeared to have
spread over the entire Lake Bolshoe Toko, with a distinct tendency of the ratio
between planktonic and benthic species to greater water depths (r 0.74, Fig. 5 and
12), due to the limited availability of light for benthic species, as reported in other
lakes (Gushulak et al., 2017;Raposeiro et al., 2018). Especially *Aulacoseira* species
were never abundant along the shallower northern and eastern shorelines. The
main difference between the two most abundant genera in the lake was that
*Pliocaenicus* showed highest abundancies in proximity to the inflow and in the
southeastern lagoon, whereas *Cyclotella* valves were more frequent in the lake
center and absent in the lagoon. There is yet little known about the new species
*Pliocaenicus bolshetokoensis (Genkal et al., 2018).* Our findings suggest factors
other than water depth (r 0.39), such as proximity (e.g. nutrient supply) to the Utuk
river and small streams, as controlling parameters for bloom intensities of this
species. *Cyclotella*, however, is restricted to stratification of the water column and



hence is more abundant at distance from the river mouth, where incoming water
causes turbulence (Rühland et al., 2003;Smol et al., 2005). *Cyclotella* is therefore
also believed to benefit from recent air temperature warming trends and will likely
increase in abundance (Paul et al., 2010). *Aulacoseira* is a heavier, rapidly-sinking
tychoplanktonic group of species requiring water turbulence to remain in the photic
zone (Rühland et al., 2008;Rühland et al., 2015), which explains the lower
abundancies in the northern and quitter zones within the lake. *Tabellaria* species are
known to occur in a planktonic lifestyle with the help of zigzag colonies and relatively
lightly silicified frustules. However, they also can appear as short-valved
populations, which are believed to represent benthic forms (Lange-Bertalot et al.,
2011;Biskaborn et al., 2013a;Krammer and Lange-Bertalot, 1986-1991). In our
study, the spatial within-lake distribution of *Tabellaria* forms in Bolhsoe Toko
indicates benthic habitats rather than planktonic lifestyle.
The most common non-planktonic species in Bolshoe Toko belong to
achnanthoid (monoraphid) genera, of which most species are epiphytic. Epiphytic
species showed a stronger negative correlation to water depth (r -0.68), than
epibenthic species (r -0.4), indicating that water plants, controlled by water
transparency, pH, water depth and nutrient status (Valiranta et al., 2011), represent
an important function in the lake ecosystem (Fig. 12). The highest amount of
achnanthoid and cymbelloid valves was found in at 400 m distance to the northern
shore at a water depth of 0.5 m.
Fragilarioid species are adapted to rapidly changing environments and thus
higher ecosystem variability (Wischnewski et al., 2011). The peak occurrences of
*Staurosira* species, which are pioneering small benthic fragilarioids (Biskaborn et al.,
2012), therefore indicates the formation of a new ecosystem habitat type in the
lagoon at the south-eastern lake basin. We assume that this basin is successively
being separated from the main basin and eventually will form a separated small
remnant lake following the example of the small "Banya" lake four kilometres from
the lagoon towards northeast (Fig. 1). High productivity of epiphytic species and low
detrital input suggested by elemental and grain-size data, together with higher
organic contents (High TOC and Br), indicate a calm sedimentological regime with
high bioproductivity. Similar neutral pH values measured in water samples from the
central basin and the lagoon (Table 1) questions pH as a main driving factor of the
*Eunotia* peak in the lagoon. However, Barinova et al. (2011) suggest 5.0-5.8 pH
range for the identified *Eunotia* species, which rather indicates that the pH values
obtained during April in 2013 are not representative for the annual average and the
specific catchment of the lagoon, which likely will differ from this point measurement.
The ice break-up during spring and transport of water from the catchment restricted
to the lagoon likely leads to milieu differences in the lagoon relative to the main
basin.





The RDA biplot of diatoms (Fig. 2) suggests that both, water depth and distance to river are important lake attributes that explain the species distibutions across the lake. Especially *Eunotia*, fragilarioids, *Tabellaria*, and also *Aulacoseira subarctica* appear more frequently at sites that are close to the Utuk river mouth (e.g. PG2113, PG2115, PG2117, PG2118). The high TOC/TN ratios in these samples illustrates the strong riverine input of allochthonous material. In the biplots, high water depth is primarily associated to *Cyclotella* species (and *Aulacoseria*), while *Aulacoseira* species tend to be additionally influenced by incoming rivers and also thrive closer to the shorelines. Areas close to river mouths are usually dominated by river taxa and species that prefer higher nutrient contents related to river input and associated early ice cover melting (Kienel and Kumke, 2002). Accordingly, the influx of diatoms from wetlands in the lake catchment is an important additional factor influencing the spatial diatom distribution (Earle et al., 1988). As compared to conductivity, water depth and nutrients, analyses on the direct relationship between temperature and diatom species is poorly understood in Yakutian lake systems (Pestryakova et al., 2018) and should be omitted.

Our RDA also shows that a high diversity of benthic, and especially epiphytic diatom species, i.e. several achnanthoid species and some naviculoid taxa plot in the opposite direction from water depth together with vegetation and the coarse grain-size fraction. Kingston et al. (1983), revealed spatial diatom variability in the Laurentian Great Lakes, where the stability of diatom assemblages increased with water depth. In shallower marginal waters of the Great Lakes, the availability of diverse habitats, including benthic and periphytic niches, leads to high species diversity. According to our data in Bolshoe Toko, the Simpson index suggests higher beta-diversities associated to increased habitat complexity (Kovalenko et al., 2012), i.e. availability of water plants and benthic substrates in shallower depths along the eastern and northern shore. The higher productivity in this area can be explained by differential catchment preferences. However, it can be assumed that due to lesser water supply rates from the small northern part of the catchment (Fig. 1), a single spot at the north eastern lake margin will likely not receive significantly higher loadings of nutrients as compared to the Utuk river coming from the igneous mountain range. Nevertheless, moraine deposits typically contain high amounts of silt and clay which can more easily be weathered and altered to fertilizing substances that are transported into the calm and shallower northern part of the basin.

The indices of crysophyte cysts and *Mallomonas* relative to diatom cells exhibited indistinct patterns in the spatial distribution but a slight tendency towards high water depths. Although crysophyte cysts commonly represent planktonic algae (Smol, 1988a), periphytic taxa are also common in boreal regions (Douglas and Smol,





1995) with cool and oligotrophic conditions (Gavin et al., 2011). *Mallomonas* was
reported as an indicator of lake eutrophication and acidification (Smol et al., 1984).
Taphonomic effects on the preservation of subfossil assemblages are generally
influenced by clastic transport mechanisms depending on the lake morphology
(Raposeiro et al., 2018). The preservation of diatom valves in Bolshoe Toko was
lowest in samples from a plateau-like feature at the central part of the lake bottom,
which indicates increased re-working associated to bottom currents and/or
increased dissolution of diatom valves due to lesser accumulation rates, and/or
increased grazing activity of herbivorous organisms (Flower and Ryves, 2009;Ryves
et al., 2001).
The spatial distribution of $\delta^{18}O_{diatom}$ from the sediment surface showed higher
$\delta^{18}O_{diatom}$ values at the deeper, south-western part of the lake with a difference of
app. 1‰ compared to lower $\delta^{18}O_{diatom}$ values in the shallower northern part. This
could be due to $\delta^{18}O_{water}$ variations, a difference in the average water temperature or
a varying species composition assuming a species-effect exists. Several studies
gave evidence that a species effect does not exist for this proxy (Bailey et al., 2014).
Additionally, the sieving step reduces the assemblage before the isotope analysis to
only a small size interval resulting in a similar species-composition. Regarding
variations in the isotope composition of the lake water surface waters sampled at
the same time in different parts of the lake show a very uniform isotopic composition
(within ±0.15‰) suggesting a well-mixed lake system and no strong seasonal
intermittency. As this is a one-time recording, slight seasonal variation between
shallower and deeper parts (for example due to evaporation) cannot be completely
excluded and could account for part of the differences in $^{18}O$. An evaporation driven
heavier water isotope composition in the shallower parts would however result in
higher $\delta^{18}O_{diatom}$ values.
The lake temperature in which the diatoms grow has an impact of ca. -0.2‰/°C
on the oxygen isotope signal (Brandriss et al., 1998;Dodd et al., 2012;Moschen et
al., 2005). Shallower areas heat up faster especially in the photic zone. The
temperature profile near to the western shoreline taken in August 2012 (Fig. 6)
shows 12°C at the surface with an average of app. 10°C in the first 15m of the water
column decreasing to app. 6°C in 30m depth. Although a spatial difference of 5°C in
the photic zone for causing a 1‰ shift is rather unlikely, this could account for part
of the variation in surface $\delta^{18}O_{diatom}$.
**5.3  Factors explaining the spatial chironomid distribution**
The RDA performed on chironomid data suggested that the most important factor
driving the spatial distribution of chironomids in the Bolshoe Toko was influence of



the tributary rivers. Influenced sites demonstrate high species richness with the
highest diversity being found at the site 2117 situated just opposite of the Utuk river
inflow and at the site PG2122, that is situated in the SE lagoon that gets water from
a small inflowing stream. Semiterrestrial taxa, like *Smittia-Parasmittia,*
*Pseudosmittia, Limnophies-Paralimnophies*, have been found only here with the
highest abundancies of 6 and 3.2% at the sites opposite of the inflowing rivers
(PG2117 and PG2122) suggesting that those taxa were transported from the dump
or marshy river deltas.
Species at lentic sites with no tributary influence were mainly controlled by water
depth. Deep profundal sites of the lake have much lower taxonomic richness of
chironomid communities. Higher taxonomic richness at site PG2118 can be
explained by an enriching riverine influence. High proportions of lotic and lotic-lentic
taxa lead to a high taxonomic similarity of this profundal site to litoral sites in the S
and SE. Similarly, in relation to temperature, sublittoral and profundal sites both
have much higher representation of the taxa characteristic of semi-warm conditions
and lower abundancies of the taxa preferring warm and cold conditions. However,
high depths of the sublittoral and profundal sites lead to the development of a poor
chironomid fauna at these sites. High distance from the shore and presumably only
weak transportation of chironomid remains of littoral fauna to the profundal zone
could be another limiting factor for diversity of chironomid communities in the
profundal.
Eastern relatively shallow littorals are inhabited by more diverse, phytophilic,
mesotrophic and partly acidophilic fauna with absence of lotic taxa, related to a less
disturbed and turbulent environments and presence of macrophytes. This fauna has
higher abundance of the semi-warm and warm taxa. The presence of meso- to
eutrophic and acidophilic taxa can be attributed to paludification of the shore zone
and decomposition of macrophytes and submerged vegetation in the shallow littoral
(Nazarova et al., 2017b).
It is still debated how spatial and local environmental processes influence the
distribution of chironomids at a small spatial scale in a lake (Luoto and Ojala,
2018;Yang et al., 2017). It is known that even within one water body the
concentration of chironomid head capsules may vary from zero to several thousand
per 1 cm$^3$ of sediments (Kalinkina and Belkina, 2018;Walker et al., 1997),
depending on the various ecological factors including the water depth, rate of
sediment accumulation, the hydrological conditions, or anthropogenic influence.
Water depth is a major driving factor of chironomid assemblages in many studies
(Ali et al., 2002;Luoto, 2012;Vemeaux and Aleya, 1998) and depth optima of several
species prove to be consequent across broad spatial scales (Nazarova et al., 2011).
Taphonomy assumes that the assemblage of chironomid remains from the deepest
zones of lake represents an assemblage of elements of profundal necrocenosis
(Hofmann, 1971) mixed with secondary components of littoral fauna transported





with in-lake hydrological and sedimentary processes into the profundal from outside.
Profundal necrocenosis therefore are supposed to include the assemblage of
remains of organism that inhabited the whole lake and are therefore the most
reliable indicators of ecological conditions in palaeoecological research (Brooks et
al., 2007).
The redeposition of littoral taxa into the profundal zone is an important factor that
affects the final composition and abundance of fossil assemblages and thus further
ecological information that can be derived from the assemblage. In small lakes,
fossil assemblages from the profundal zone quite adequately reflect the fauna of the
entire water body (Brooks and Birks, 2001;Walker and Mathewes, 1990). Our
findings support the hypothesis that in large lakes taphonomy of chironomid
communities seems to be more complex (Yang et al., 2017;Árva et al., 2015).


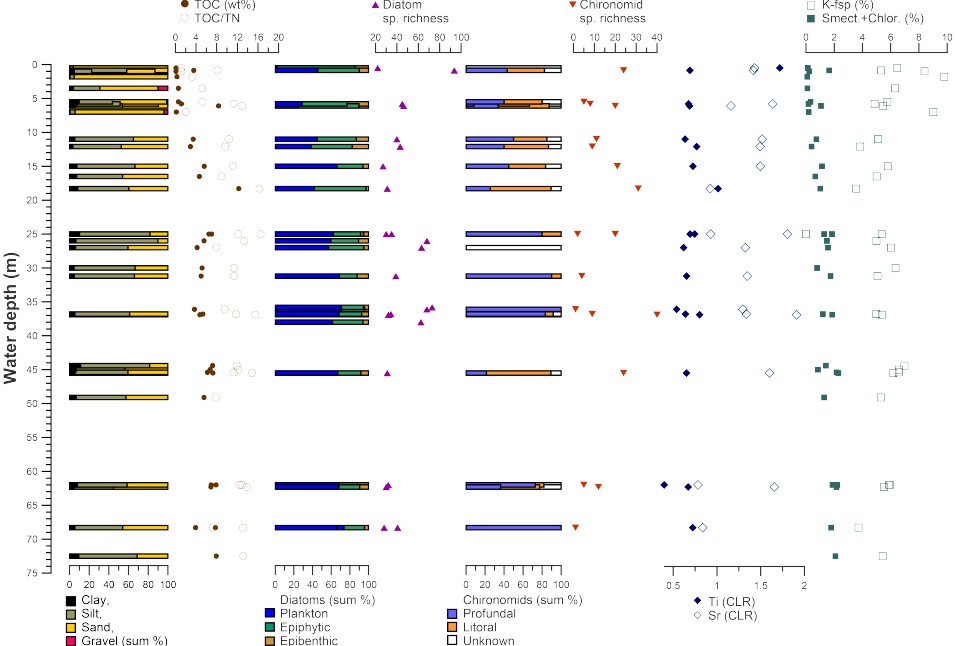

**Fig. 12** Distribution grain size, organic carbon and nitrogen indices, diatom and chironomid parameters, and
selected elements and minerals in dependence to lake water depth.

**5.4  Lake Bolshoe Toko as a site for palaeoclimate reconstructions**
Compared to small lowland lakes of Central and Northern Yakutia sedimentary
processes and diatoms assemblages are quite different in Bolshoe Toko. One
obvious reason for this is the lack of thaw slumps, subsidence, and other permafrost
related phenomena (Biskaborn et al., 2013b) that are typical for shallow thermokarst



lake settings across northern permafrost regions (Biskaborn et al., 2016;Bouchard
et al., 2016;Biskaborn et al., 2012;Schleusner et al., 2015;Biskaborn et al.,
2013a;Subetto et al., 2017;Biskaborn et al., 2013b).

The mineral composition in Bolshoe Toko was mainly influenced by the Utuk river
and only the samples in extremely shallow areas were influenced by direct shoreline
input. The grain-size signal was influenced by dissolution effects associated to
organic matter and by in situ growth of diatom valves. The coarser fractions varied
parallel to minerogenic compositions and water depth. Accordingly, the grain-size
distribution originated from multiple processes and should only be considered as an
environmental proxy in combination with biotic indicators.

Diatoms were distributed mainly according to their preferential habitats. Aside of
the spatial habitat conditions associated to the basin morphology, an additional
principal determinant of shifting diatom assemblages in cold environments is the
annual duration of lake ice-cover (Keatley et al., 2008;Smol, 1988b). Heavily
silicified planktonic diatoms (e.g. *Aulacoseira*) cannot survive below the lake ice-
cover because of the absence of wind-driven water turbulence. Nevertheless,
planktonic and benthic diatom species have strategies to survive in ice-covered
lakes, growing in benthic mode or attached to the bottom of the ice-cover (D'souza,
2012). Hence, in many lakes, the presence or absence of the ice-cover influences
blooms of different species which can result in changes of both the species
distribution and the ratio of planktonic to benthic diatoms (Wang et al., 2012a).

The applicability of chironomids for temperature reconstructions reveals clear
spatial constraints. 22% of the taxa found in sites with riverine influence are absent
or rare from the FE mean July chironomid-based temperature inference model
(Nazarova et al., 2015), whereas the sum of the taxa that are rare and absent in FE
data set is much lower in the central and northern littoral, sublittoral and profundal
part of the lake (Fig. 4). However, low taxonomic richness of the profundal zone as
well hampers palaeoclimatic inferences. The number of chironomid head capsules
were generally lower here in relation to littoral sites. The highest taxonomic richness
in areas influenced by lake tributaries can be explained not only by a taxonomic
enrichment from the lake catchment but as well by more favorable oxygen and
nutrient conditions.

The applicability of $\delta^{18}O_{diatom}$ as a proxy of past hydroclimate conditions at
Bolshoe Toko is generally facilitated by the fact that the main controls influencing on
$\delta^{18}O_{diatom}$ in a lake are (1) the lake water temperature ($T_{lake}$) and (2) lake water
isotope composition ($\delta^{18}O_{lake}$) (Dodd and Sharp, 2010;Leng and Barker,
2006;Labeyrie, 1974;Leclerc and Labeyrie, 1987). The fractionation between lake
water and biogenic opal can be calculated when comparing $\delta^{18}O_{lake}$ (mean: −18.7‰)
with recent surface sediments of Bolshoe Toko lake and their respective mean
$\delta^{18}O_{diatom}$ (of +22.8‰) using this isotope fractionation correlation between fossil





diatom silica and water as determined by Leclerc and Labeyrie (1987). The mean
$T_{lake}$ can be estimated to ca. 6°C for the photic zone/diatom bloom. This estimate is
at the lower end of summer temperatures between 4.8 and 12°C. The
corresponding derived mean isotope fractionation factor for the system diatom
silica–water α = 1.0424 is matching the fractionation factor for fossil sediments
proposed by Juillet-Leclerc and Labeyrie (1987) well ($\alpha_{(silica-water)}$ = 1.0432).
As the lake water isotope composition ($\delta^{18}O_{lake}$) is further governed by
precipitation intermittency in the catchment, $\delta^{18}O_{diatom}$ will react on changing isotopic
composition in precipitation produced along the pathway of air masses to the study
area, seasonality patterns and influenced by air temperature changes. Despite the
observed slight spatial shifts in the surface samples $\delta^{18}O_{diatom}$ changes over time at
a single site will yield insights into the air temperature and precipitation history of the
area.
Positive feedback mechanisms were previously described between benthic algae
and chironomid larvae in benthic ecosystems (Herren et al., 2017). Chironomids in
Bolshoe Toko, however, showed less significant correlations with benthic diatom
species, but weak correlations with planktonic species and lake attributes
associated to benthic habitats and water depth, highlighting the potential of
chironomids for independent water depth and temperature reconstruction in future
sediment core studies (Nazarova et al., 2011).
High correlation coefficients between organic carbon and *Pliocaenicus*
*bolshetokoensis* (0.66), and silt (0.65) suggest that the accumulation of organic
matter, and the intermediate grain-size fraction, is to a certain degree controlled by
the productivity of siliceous microalgae (Biskaborn et al., 2012). A strong
contribution of plankton indicates that TOC/TN ratios can provide insights in the
relative influx between land and water plants (Meyers and Teranes, 2002). The
relatively weak correlation between TOC/TN ratios and water depth (0.51 r),
demonstrates the accuracy limits of TOC/TN as a proxy for relative lake level
changes. This is caused by transport and accumulation of allochthonous organic
matter in proximity to the Utuk river. Furthermore, correlations between TOC/TN and
TOC, as well as negative correlations with grain size indicators suggest diagenetic
alteration (i.e. loss) of nitrogen in the surface sediments (Galman et al., 2008).
The distinct difference between two samples along the subaquatic slope near the
western shore (diatoms, minerals, organics) indicates redistribution of sediment.
Downslope transport of surface layers over the time could lead to redistribution of
old material into the deepest parts of the basin. Due to higher accumulation rates, a
sediment core from the deepest part of the basin would potentially provide a higher
temporal resolution, but also a higher risk of repositioned sediment layers. On top of
redistribution processes, hump-shaped relations between lake depth and species
diversity observed in other studies suggest that the total subfossil species



assemblages is better represented at intermediate depths than at the maximum depth (Raposeiro et al., 2018). A coring position at intermediate depth in the northern shallower and sedimentologically calm part of the basin would enable the tracking of different intensities of river influence and glacial activity using sediment-geochemical indicators and offers greater chances of undisturbed successions of bioindicator time series.

## 6 Conclusions

Our study on the within-lake variance of environmental indicator data and its attribution to habitat factors improves the understanding of lake-internal filters between environmental forcing and the resulting sediment parameters of Lake Bolshoe Toko and comparable boreal, cold, and deep lakes. We found that the spatial variabilities of biotic ecosystem components are mainly explained by static habitat preferences as water depth and river distance. Abiotic sediment features are not symmetrically distributed in the basin but vary along restricted areas of differential environmental forcings (e.g. river input, rocky shore, steep shore, shallow shore). They depend, in addition to water depth and riverine activity, to multiple interacting factor, such as catchment characteristics, geochemical sediment diagenesis and hydrochemical dynamics. Our main findings can be highlighted as follows:

- The lake water of Bolshoe Toko can be characterized as Ca-Mg-HCO3-Type water. It is well saturated in $O_2$, neutral to slightly acidic, showing a low conductivity and corresponding ion concentrations suggesting unpolluted freshwater conditions. Lake Bolshoe Toko is likely a cold, polymictic, oligotrophic, open through-flow lake system and due to all stated aspects regarded as an undisturbed ecosystem.
- Water depth is a strong factor explaining the variability of diatoms and chironomids. The proportions of planktonic to benthic diatoms as well as profundal to littoral chironomids serve as a reliable lake level proxy.
- The diatom assemblage is dominated by planktonic species, i.e. *Pliocaenicus bolshetokoensis*, which is unique for this lake, and more common plankton such as *Cyclotella* and *Aulacoseira*, as well as non-planktonic taxa, such as *Achnanthidium*. Diatom species richness and diversity is higher in surface sediments in the northern part of the basin, associated to shallower waters and the availability of benthic and periphytic niches.
- The $\delta^{18}O_{diatom}$ values (22.8±0.6‰) show slight spatial variations with higher values in the deeper south-western part of the lake probably related to water temperature differences in the photic zone during the main diatom bloom.

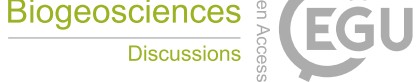



The silica–water fractionation is suitable for further downcore investigations for assessing paleo-hydrological information and potential air-temperature changes in the region.

- The water of Bolshoe Toko is well mixed and does not show significant isotopic stratification apart from lake ice-cover formation where thermal stratification prevents mixing. The isotopic lake water composition ($\delta^{18}O$ = 18.2 ± 0.2 ‰) correspond with the GMWL and do not show evaporative enrichment. Both isotopic and hydrochemical data indicate atmospheric precipitation (and meltwater run-off) as the main water source. Accordingly, $\delta^{18}O_{lakewater}$ is directly linked to $\delta^{18}O_{precipitation}$.

- The highest amount of the chironomid taxa underrepresented in the FE training set used for palaeoclimate inference was found close to the Utuk river and at southern littoral and profundal sites. Poor chironomid communities from the deep profundal zone would also hamper palaeoclimate reconstruction. Cold-stenotherm chironomid taxa were influenced by river proximity while taxa preferring warm conditions were more frequent at shallow littorals of the lake.

- Weak negative correlation between mean grain size and water depth is explain by end-members revealing influences of river input and diatom valves in the grain-size composition.

- Observed TOC values (mean 4.9 %) and TOC/TN ratios indicate strong allochthonous supply of organic matter from the Utuk river. $\delta^{13}C$ (mean -26.8 ‰) indicate dominance of $C_3$ plants and phytoplankton in the bulk organic matter fraction.

- Elemental (XRF) data and mineral (XRD) distribution is influenced by the methamorphic lithology of the Stanovoy mountain range. Ratios of minerals relative to quartz decrease from the Utuk river towards the northern lake basin. Ti correlates well with mean grain size. There is no clear pattern in Mn/Fe ratios, due to mixture of allochthonous elements and differential intensities of redox processes in the lake basin.

## Data Availability

All data used in this study will be available online at PANGAEA.

## Supplement

The supplementary material related to this study will be available online at Copernicus.



**Author contributions**

BKB conceiced the study concept, conducted or led the laboratory analyses and led the writing of the manuscript. LN conducted statistical analyses and contributed with ecological chironomid expertise. LAP led the Russian team during field work and contributed with ecological diatom expertise. LS conducted chironomid analysis. KF conducted diatom analyses. HM conducted water chemistry analyses. BC analysed diatom opal oxygen isotopes. SV conducted the XRF analysis. RG and EZ retrieved surface samples during field work and helped with translation of Russian literature and geographical expertise of the study area. RW conducted grain-size analyses including end-member modelling. GS conducted XRD analyses. BD was the leader of German expedition team and contributed with sedimentological expertise.

**Competing interests**

The authors declare that they have no conflict of interest.

**Acknowledgements**

The expedition Yakutia 2013 was financed and conducted by the Alfred Wegener Institute Helmholtz Centre for Polar and Marine Research in Potsdam, Germany in collaboration with the North Eastern Federal University in Yakutsk, Russia. Parts of the study was financed by the Federal Ministry of Education and Research (BMBF) in the PALMOD project (#01LP1510D) and grant #5.2711.2017/4.6, the Russian Foundation for Basic Research (RFBR grant #18-45-140053 r_a), and the Project of the North-Eastern Federal University (Regulation SMK-P-1/2-242-17 ver. 2.0, order No. 494-OD), Russian Science Foundation (#16-17-10118), and Deutsche Forschungsgemeinschaft DFG (#NA 760/5-1 and #DI 655/9-1). We thank Almut Dressler and Clara Biskaborn for help with diatom microscopy and Thomas Löffler for help with mineral analyses. We thank the anonymous reviewers for their voluntary efforts to assure the quality of this study.

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
