# Peer review of "Spatial distribution of environmental indicators in surface"

_Biogeosciences, 2019_

## Referee Comment (RC1) · 17 Jun 2019

This manuscript presents an extensive, multiproxy investigation of Bolshoe Toko, a large lake in Yakutia (northern Russia). In this study, the authors measured and analyzed a series of environmental indicators from the lake in order to better understand how it functions with the goal of making an informed decision on the best regions to retrieve sediment cores in order to provide the best possible sedimentary sequences to infer past environmental changes in the lake, its catchment and the region. Abiotic (sedimentological, isotopic, etc) and biotic (diatom, chironomid) components of the system were investigated, allowing for an in-depth analysis of the current state of the

lake.

I think that this type of investigation should be standard when large lakes are targetted for paleoenvironmental studies. This team of researchers has done an excellent job of establishing current links between environmental variables and their effects on the various abiotic and biotic components of the Bolshoe Toko system. With their holistic and regional-scale understanding of the current lake system, they will be very well prepared and equipped to analyze the data from a long sediment core.

The text is well-written and easy to understand, albeit with some small grammatical errors that can easily be fixed (see specific comments below). The figures are clear and eye-pleasing.

There are two things that the authors could have included in their investigation that could add even more useful information to refine the interpretation of future results obtained from sediment cores: 1) lake residence time and 2) assessment of a possible reservoir effect/input of old carbon from the catchment to the lake basin (dating of surface sediments provides a straightforward indication of the presence of these). Perhaps these can be mentioned in the text as possible ways to improve this type of preliminary study in future, especially in northern regions, where obtaining reliable chronologies can be challenging.

The manuscript is long due to the high number of components investigated. I suggest putting the two tables in Supplementary Materials in order to shorten the main text. I would also like to see the diatom (and chrysophyte) and chironomid data in Supplementary Materials. The authors should make an effort to be extra concise in their wording.

Specific comments: Figure 6B: Adjust the axes in this figure; shorter axes will allow a better view of the variability in the data.

Please replace the term "fossil" when referring to biotic components found in the sediment. This is an incorrect use of the word. You can either use sedimentary remains or sub-fossils, for example.

Sometimes, the references are underlined. Please make sure to remove this.

The Resuts section should be written in past tense. Make sure that this is the case (there are some sentences written in the present tense).

The Discussion should begin with a stement of your main finding(s).

Line 85: remove the "s" in content Line 168: length instead of diameter Line 173: remove "The" at the beginning of the sentence Line 183: replace "northern direction' with North Line 213: please include the years on which the mean temperatures are calculated Lines 285-292: this information belongs in the Results section Line 291: before and prior mean the same thing; remove one Line 403: information is missing in this sentence (where or what was the data derived from?) Line 647: remove the capital "S" in sand Line 870: what does "quitter" mean? Did you mean quieter (less turbulent)? Line 894: remos the "s" in content Lines 957-958: You mention "several studies" but you only cite one. Line 1043: remove the "s" from diatom Line 1256: Aside from (not of) Line 1290: replace "is matching" with matches

―――――――――――――――――――

---

## Referee Comment (RC2) · Anson Mackay (Referee) · 25 Jun 2019

General comments

This is a comprehensive study that looks at spatial variation of a number of biological, sedimentological, and isotopic indicators in a pristine lake (Lake Bolshoe Toko) in southern Siberia. Ultimately, the data will be used to inform on potential coring locations for long records, and to aid with palaeoenvironmental interpretations. The study has a number of strengths. Few studies critically consider the potential impact of coring location on the palaeolimnological indicators in the sedimentary record (although there are notable exceptions, see below). What makes this study stand out is
the range of potential proxies / indicators that are covered. However, this also leads to potential weaknesses in the manuscript: (i) almost everything possible has been done, but (except for diatoms and chironomids) without the required reflection as to their palaeoenvironmental merits. This can be easily accommodated however. (ii) was a robust sampling strategy taken for each indicator – see comment below; (iii) I didn't see consideration of what role spatial autocorrelation may have played in the observed statistical relationships, which is likely to be important due to location of samples from the same lake. Given that this is a very interesting study, I hope that my comments help the authors to highlight more clearly which relationships might really be important.

I have made quite a few comments and spelling / grammatical corrections on the PDF. But I highlight some of the more important aspects in the specific comments below.

Specific Comments:

In the final line of the abstract, I don't know of any lakes that are not suitable for multi-proxy analyses. So I wonder if the abstract ought not end with a statement as to where the optimal coring location(s) is/are, and why.

Introduction: An assessment of lake heterogeneity has been tackled by many previous studies, e.g. I know of three done in our research group alone (e.g. Zhao et al. 2006). So a deeper consideration of what is being done here is warranted in light of studies that have gone before.

Lines 110-114: in terms of the diatom isotopes, one should also know catchment processes, such as the isotopic composition of inflows during both summer and winter. And I would suggest dissolution (Smith et al. 2016), which I'll come back to below.

Lines 147-151: great to see testable hypotheses here, but they are very vague. As phrased, you are almost certainly to find something that correlates, and we kind of know already from many studies that habitat properties will play a role in influencing bioindicators (e.g. coring from shallow or deep parts of lakes). It might be more useful to hypothesize about Lake Bolshoe Toko specifically?

Line 163: which is meant by a stressor in a non-impacted lake?

Study Site: Line 190-191: Would be useful to give recent productivity figures

Material and Methods:

These are all largely fine. What would be useful to know is (i) why not all 42 core tops were analysed for each proxy, ie give a rationale for looking at numbers of samples detailed for each indicator; (ii) with each proxy, state why it is being measured. For example, what are 18Odiatom values expected to reveal about the environment? Otherwise, there is the danger that the study becomes a bit descriptive.

Line 270: alkalinity?

Lines 410-412: dissolution and concentration calculations are not statistical analyses – more to section above.

Line 426: PCA may capture more variance, but do the data have a horseshoe shape when sample scores only are plotted for Axis 1 and 2? DCA is mainly done to get rid of this artefact. If this is present, then PCA is not appropriate.

Line 442-443: just a comment: given likely collinearity of many of the explanatory variables, p = 0.05 is quite high, and therefore significance easy to achieve. Might be better to consider a more robust p value of e.g. 0.005 or or even 0.001 to determine what is important (Colquhoun 2015).

Line 452: define Hill's N2 here

Line 460-461 – is this a form of PCA?

Line 465: given likely very strong autocorrelation in the spatial datasets, p = 0.05 is quite high, and therefore significance easy to achieve. Might be better to consider a more robust p value of e.g. 0.005 or or even 0.001 to determine what is important (Colquhoun 2015). Or test significance once spatial autocorrelation has been taken into account.

Fig 2 & 3: It really would be better to show the distribution of the biological proxies using indirect ordination first of all, and then show the more advanced ordination (RDA). A biplot of samples ordered by indirect ordination, just on the basis of biological composition, can be immensely informative.

In the RDA, did you test the explanatory variables for normality before analyses? Which ones had to be transformed before the analyses? I can't see this information anywhere, yet this is very important.

Fig 5: In one sense these analyses are fine. But as autocorrelation will be so high here, I think a more robust p values is warranted, else there is a danger of getting lots of Type 1 errors

Line 534-535: Does it matter that virtually none of the samples actually lie on the GMWL? The lagoon shows signs of evaporation, but almost all the samples are below the GMWL, so are the isotopes influenced by ocean sources with high humidity?

Results: Table 1: It might be useful to show these relationships in a PCA, with variables standardised to take account of different units... Only include variables above detection limits.

Line 637: What does this index tell us?

Fig 7- 10 With all of these figures, I'm not convinced by the use of the green - red scale to represent low to high; how are scales chosen? Why do some maps have purple? It would be good to know how objective the choice of scales was.

Line 994-995: If this is the rationale for including Chrysophytes, then perhaps state this earlier. Are there any conclusions from their distribution in the lake?

Line 976-977: the authors should also consider the role that dissolution can play here, especially for younger material, e.g. see Smith et al. 2016. DOI: 10.1002/rcm.7446

Line 1061-1062: This need not be the case. For example, in Lake Baikal, Aulacoseira species do very well under the ice, and I'm sure this could be the same for other non-shallow lakes. Eg see Jewson et al. 2009

Sections 5.4 and the conclusions are all good, but it might be also good to provide a concluding statement about potential for optimal coring location

References used in the review:

Colquhoun, D. (2015) An investigation of the false discovery rate and the misinterpreptation of p-values. Royal Society Open Science. http://rsos.royalsocietypublishing.org/content/1/3/140216

Jewson, D.H., Granin, N.G., Zhdarnov, A.A., Gnatovsky, R.Y. (2009) Effect of snow depth on under-ice irradiance and growth of Aulacoseira baicalensis in Lake Baikal. Aquatic Ecology, 43, 673–679.

Smith, A.C., Leng, M.J., Swann, G.E.A., Barker, P., Mackay, A.W., Ryves, D.B., Sloane, H., Chenery, S.R.N., Hems, M. (2016) An experiment to assess the effects of diatom dissolution on oxygen isotope ratios. Rapid Communications of Mass Spectrometry 30, 293-300. DOI: 10.1002/rcm.7446

Zhao, Y., Sayer, C.D., Birks, H.H., Peglar, S.M. & Hughes, M. (2006) Spatial representation of aquatic vegetation by macrofossil and pollen remains in a small and shallow lake. Journal of Paleolimnology 35, 335-350.

Please also note the supplement to this comment: https://www.biogeosciences-discuss.net/bg-2019-146/bg-2019-146-RC2-supplement.pdf
* * *
[Figure]

**Supplement:**

[revised manuscript text omitted]

---

## Author Comment (AC1) · 12 Jul 2019

Dear Dr Saulnier-Talbot

Thank you very much for your review of our manuscript. We are grateful for the positive feedback. This motivates us a lot. We also acknowledge your suggestions for further analyses and detailed comments that helped to improve the quality and readability of the manuscript. We already performed radiocarbon analyses of the surface sediments and can provide basic assessments of a reservoir effect. We also can work on an assessment regarding the lake residence time. We plan upload all data used in this manuscript to PANGAEA and we can also include more graphs on bioindicators in the

supplementary material

We would like to prepare detailed revision notes and a point-to-point answer to each of your comments as soon as possible.

We highly acknowledge your efforts. With kind regards, Boris Biskaborn

---

## Author Comment (AC2) · 12 Jul 2019

Dear Dr Mackay

Thank you very much for your comprehensive review of our manuscript. We agree with your suggestions and we believe that we can solve most of the problems you are pointing at. Your detailed comments on the statistics applied are very valuable, thank you very much. It will help to focus this paper more on the specific characteristic of Bolshoe Toko including conclusions for environmental implications and core locations. Yes we can adjust our significance test to p 0.001. We are aware that this study has a high descriptive component, which was part of the study design given that we plan

to submit follow-up papers on long sediment core material from this lake. The usual dimension of one manuscript suggests to rather split the downcore analyses from the spatial proxy variability, so that we can refer to the results in this manuscript when discussing the proxy variability over time in our next manuscripts.

We would like to prepare detailed revision notes and a point-to-point answer to each of your comments.

We highly acknowledge your great efforts reviewing our manuscript and we are grateful for the additional literature you provided.

With kind regards, Boris Biskaborn

---

## Author Response (AR1)

**Revision Notes to manuscript bg-2019-146:** *Spatial distribution of environmental indicators in surface sediments of Lake Bolshoe Toko, Yakutia, Russia*

*19 August 2019,* Biskaborn et al.

**Editors decision**

*We have now received two reviews of your manuscripts. Both reviewers are reasonably positive about the work, but they also have made several important comments/observations that you must consider. Referee 2 in particular has offered a number of important comments both concerning the substance and presentation that you need to take into account while revising the manuscript. Kindly make sure that you address each and every point raised by this referee carefully because the revision will again be sent to him.*

*Looking forward to receiving the revised manuscript and with kind regards*
*Sincerely*
*Wajih Naqvi*

> (Our answers indented and marked in blue)
>
> Dear Dr. Naqvi,
>
> Thank you very much for the possibility to resubmit our manuscript. We revised the manuscript carefully following all comments of both reviewers, including statistical analyses, figures and tables, the ESM, and the presentation of our study in the text. We documented all changes in the revision notes provided in the following point-to-point answers.
>
> **Please note:** We added Hannah Bailey as co-author to this manuscript because she contributed to the original laboratory work including analysis of diatom oxygen isotopes significantly, and has now further contributed text and edits to the revised manuscript. She was suggested as a reviewer to this manuscript, but it turned out that she is much better suited as a co-author - regarding ethic

guidelines of the Deutsche Forschungsgemeinschaft (DFG). Thank you very much for understanding.

With best regards on behalf of all authors,
Boris Biskaborn

**Reviewer #1 Émilie Saulnier-Talbot and our answers**

*This manuscript presents an extensive, multiproxy investigation of Bolshoe Toko, a large lake in Yakutia (northern Russia). In this study, the authors measured and analyzed a series of environmental indicators from the lake in order to better understand how it functions with the goal of making an informed decision on the best regions to retrieve sediment cores in order to provide the best possible sedimentary sequences to infer past environmental changes in the lake, its catchment and the region. Abiotic (sedimentological, isotopic, etc) and biotic (diatom, chironomid) components of the system were investigated, allowing for an in-depth analysis of the current state of the lake.*

*I think that this type of investigation should be standard when large lakes are targetted for paleoenvironmental studies. This team of researchers has done an excellent job of establishing current links between environmental variables and their effects on the various abiotic and biotic components of the Bolshoe Toko system. With their holistic and regional-scale understanding of the current lake system, they will be very well prepared and equipped to analyze the data from a long sediment core.*

*The text is well-written and easy to understand, albeit with some small grammatical errors that can easily be fixed (see specific comments below). The figures are clear and eye-pleasing.*

Dear Dr Saulnier-Talbot

Thank you very much for your review of our manuscript. We are grateful for the positive feedback. This motivates us a lot. We also acknowledge your suggestions for further analyses and detailed comments that helped to improve the quality and readability of the manuscript. We already have radiocarbon analyses of the surface sediments and can provide basic assessments of a reservoir effect. We also can provide an assessment regarding the lake residence time. We will prepare detailed revision notes and a point-to-point answer to each of your comments.

We would like to appreciate your efforts by mentioning your name in the acknowledgements. If you don't want your name to appear, please inform the first author as soon as possible.

*There are two things that the authors could have included in their investigation that could add even more useful information to refine the interpretation of future results obtained from sediment cores: 1) lake residence time and 2) assessment of a possible reservoir effect/input of old carbon from the catchment to the lake basin (dating of sur- face sediments provides a straightforward indication of the presence of these). Perhaps these can be mentioned in the text as possible ways to improve this type of preliminary study in future, especially in northern regions, where obtaining reliable chronologies can be challenging.*

To 1) lake residence time:
We agree that this is a good idea. The lake residence time is not easy to assess in sufficient accuracy without having necessary isotope and tracer measurements done. What we can do is to calculate the volume of the water body and catchment area and to compare this with the average precipitation at the closest meteorological weather station. We already started to work on a second paper dealing with historical changes of Bolshoe Toko, taking into account data of the nearby weather station Toko-RS available at NOAA and short core bioindicator data. Our estimation of lake residence resulted in about 5-6 years. Given that the calculation based on only meteorological and satellite data needs some extra figures, equations and description of the hypothetical assumptions necessary to do this, but the space available in the manuscript in hand is already at its limits, we would prefer to include this topic in the follow-up article. We would be happy if you agree with this.

To 2) radiocarbon reservoir effect:
We fully agree with this. Actually, this was planned for the next paper in which we will show the chronology of long sediment cores from this lake. However, we already can show the radiocarbon results of the surface sediments. We indeed use them to assess the reservoir effect that is likely caused by the input of old organic carbon. The issue with this is that this input is likely not constant over time, but there is no consistent methodology how to assess a potential change for different periods and thus the chronology of sediment cores usually suffers. However, the surface samples suggest that the accumulation of radiocarbon was lower than in purely fresh organic material and hence indicate input of old $^{14}$C. This is supported by $^{210}$Pb and $^{137}$Cs measurements by Appleby's Lab at Liverpool University showing that the surface sediments were deposited starting in AD 2007 at 0.5 cm until AD 2013 at 0 cm, which was the year of sample collection. However, we have to publish detailed chronology data along with sediment core data and thus opt to keep results that belong to downcore information out of the manuscript in hand. Accordingly we added in the methods, results, and conclusions as follows:

**Methods**: "Radiocarbon dating of two bulk sediment surface sample from short cores, each ranging from 0-0.5 cm depth below the sediment surface, was performed in the Poznan Radiocarbon Laboratory on the soluble (SOL) fraction using an Accelerator Mass Spectrometer."

**Results**: "Radiocarbon dating of surface sample at site PG2139 (0-0.5 cm) indicates an age of 720 ± 30 $^{14}$C yrs BP (Lab-ID: Poz-105350, NaOH-SOL), while PG2207 (0-0.5 cm) suggests 1790 ± 130 $^{14}$C yrs BP (Lab-ID: Poz-105355, NaOH-SOL. Considering that the carbon concentration dissolved in sample PG2207 was too low (0.03 mgC), we use sample PG2139 as an estimated reservoir effect to the lake caused by the input of old carbon. Given that a hypothetical sediment surface is just a momentum only collectable as a range of past surfaces and there was more time available for radioactive decay at 0.5 cm depth than at 0 cm, the actual reservoir effect will be a little bit lower and should be confirmed by $^{210}$Pb and $^{137}$Cs measurements of downcore material before establishing an age depth model for sediment cores."

**Conclusions**: "Radiocarbon dating suggests that there is a reservoir effect caused by input of old organic carbon by max. 720 ± 30 14C yrs BP."

*The manuscript is long due to the high number of components investigated. I suggest putting the two tables in Supplementary Materials in order to shorten the main text. I would also like to see the diatom (and chrysophyte) and chironomid data in Sup- plementary Materials.*

Yes, this is true, we used as many as usually applied palaeo proxies on the surface samples and the description of the methods and results is therefore necessarily extensive. We shifted the two tables as well as diatom and chironomid data in the supplementary material. All data will also be available online at Pangaea as soon as they get accepted for publication in Biogeosciences.

*The authors should make an effort to be extra concise in their wording.*

We agree and have condensed the text where possible. The entire text was also English proof read again by the English native speakers in our author group.

*Specific comments: Figure 6B: Adjust the axes in this figure; shorter axes will allow a better view of the variability in the data.*

Yes. We adjusted the axes of the d18O graph according to the range of data.

*Please replace the term "fossil" when referring to biotic components found in the sediment. This is an incorrect use of the word. You can either use sedimentary remains or sub-fossils, for example.*

Yes. We agree and changed each mention of "fossils" accordingly.

[Figure]

*Sometimes, the references are underlined. Please make sure to remove this.*

This was because we are using Endnote and there is some technical problem with an older version. It can easily be removed when removing the matches between citations and references in the MS Word environment. We would like do this together with the type setting team of Biogeosciences at a later stage of the peer review process.

*The Results section should be written in past tense. Make sure that this is the case (there are some sentences written in the present tense).*

Yes, we have now checked and amended all the manuscript text to be consistent in tense.

*The Discussion should begin with a statement of your main finding(s).*

OK. We added at the beginning of our discussion as follows: "Sediment-geochemical and physical properties of the uppermost surface of the sediment basin in Bolshoe Toko are spatially variable. Physical properties of particles within the surface sediments depend chiefly on transportation processes …"

*Line 85: remove the "s" in content Line 168: length instead of diameter Line 173: remove "The" at the beginning of the sentence Line 183: replace "northern direction' with North Line 213: please include the years on which the mean temperatures are calculated Line 291: before and prior mean the same thing; remove one Line 403: information is missing in this sentence (where or what was the data derived from?) Line 647: remove the capital "S" in sand Line 870: what does "quitter" mean? Did you mean quieter (less turbulent)? Line 894: remos the "s" in content Lines 957-958: You mention "several studies" but you only cite one. Line 1043: remove the "s" from diatom Line 1256: Aside from (not of) Line 1290: replace "is matching" with matches*

All detailed comments and gramma corrections you gave above were highly appreciated and we agreed with each and revised the wording carefully following your recommendation. Thank you very much.

*Lines 285-292: this information belongs in the Results section*

In this case we would prefer to keep this in the method section, because it describes how well the XRF spectra could get modelled into elements. Some papers do not describe this at all, but it is important for comparison with other lake systems, this is why we include it. The results section are already so extensive, that allowing these details to be seen as methodological description of the measurements would preserve the readability of the manuscript.

[Figure]

[Figure]

Thank you very much. Your effort on reviewing our manuscript helped significantly to assure the quality of this study. We additionally fine-tuned the English in the entire manuscript again and marked all additional edits in blue.

With kind regards,
Boris Biskaborn et al.

**Reviewer #2 Anson Mackay and our answers**

*General comments*
*This is a comprehensive study that looks at spatial variation of a number of biological, sedimentological, and isotopic indicators in a pristine lake (Lake Bolshoe Toko) in southern Siberia. Ultimately, the data will be used to inform on potential coring locations for long records, and to aid with palaeoenvironmental interpretations. The study has a number of strengths. Few studies critically consider the potential impact of coring location on the palaeolimnological indicators in the sedimentary record (although there are notable exceptions, see below). What makes this study stand out is the range of potential proxies / indicators that are covered. However, this also leads to potential weaknesses in the manuscript: (i) almost everything possible has been done, but (except for diatoms and chironomids) without the required reflection as to their palaeoenvironmental merits. This can be easily accommodated however. (ii) was a robust sampling strategy taken for each indicator – see comment below; (iii) I didn't see consideration of what role spatial autocorrelation may have played in the observed statistical relationships, which is likely to be important due to location of samples from the same lake. Given that this is a very interesting study, I hope that my comments help the authors to highlight more clearly which relationships might really be important. I have made quite a few comments and spelling / grammatical corrections on the PDF. But I highlight some of the more important aspects in the specific comments below.*

Dear Dr Mackay

Thank you very much for your comprehensive review of our manuscript. We agree with your suggestions and solved the problems you pointed at. We are aware that this study has a high descriptive component, which was part of the study design given that we plan to submit follow-up papers on long sediment core material from this lake. The usual dimension of one manuscript suggests to rather split the

downcore analyses from the spatial proxy variability, so that we can refer to the results in this manuscript when discussing the proxy variability over time in our next manuscripts. Your detailed comments on the statistics applied helped to asure the quality of this paper. We amplified our focus more on the specific characteristic of Bolshoe Toko including conclusions for environmental implications and core locations. We prepared detailed revision notes and a point-to-point answer to each of your comments below.

To I) Thank you for the hint. We reworded the introduction of the proxies in the method sections accordingly, added to our discussion and expanded our conclusions.

To II) Please see our detailed answer below – the sample strategy was constrained by financial issues and due to the remote location of the lake and limited amount of surface sample material.

To III) Good point. We now performed an autocorrelation analysis using sample site coordinates (R package "ape", Moran's I autocorrelation coefficient displayed as p values). This yielded interesting insights in the role of bio-process to local phenomena, e.g. reproduction, leading to spatial autocorrelation in the data sets.

We would like to appreciate your efforts by mentioning your name in the acknowledgements. If you don't want your name to appear, please inform the first author as soon as possible.

*Specific Comments:*
*In the final line of the abstract, I don't know of any lakes that are not suitable for multiproxy analyses. So I wonder if the abstract ought not end with a statement as to where the optimal coring location(s) is/are, and why.*

We agree and reworded as follows:

"We conclude that the lake represents a valuable archive for multiproxy environmental reconstruction based on diatoms (including oxygen isotopes), chironomids and sediment-geochemical parameters. Our analyses suggest preferably two correlated coring locations at intermediate depth in the northern basin and the deep part in the central basin, to account for representative bioindicator distributions and higher temporal resolution, respectively."

[Figure]

*Introduction:*

*An assessment of lake heterogeneity has been tackled by many previous studies, e.g. I know of three done in our research group alone (e.g. Zhao et al. 2006). So a deeper consideration of what is being done here is warranted in light of studies that have gone before.*

We agree and included the following sentences in the introduction citing this paper: "As previous studies described, pollen distribution in lake sediments are less influenced by lake zonation than aquatic communities (Zhao et al., 2006). Accordingly, our study does not consider spatial pollen distributions."

It might, however, still be worthy to test, whether or not pollen distributions in deep and larger lakes such as Bolshoe Toko are spatially fractionated, because Zhao et al. describe a very small and shallow lake while Bolshoe Toko is large and the deepest lake in Yakutia. However, this is not something to be tested yet in the manuscript in hand. Thank you in any case for the good comment.

*Lines 110-114: in terms of the diatom isotopes, one should also know catchment processes, such as the isotopic composition of inflows during both summer and winter.*

We agree, and now acknowledge this as good as possible , given the availability of samples, on Line 539: ""All samples were positioned close to the Global Meteoric Water Line (GMWL, Fig. 6) indicating negligible evaporative effects on lake water isotope composition, and a dominant influence of meteoric inputs both directly (i.e., precipitation) and indirectly (i.e., river inflows).. The Local Meteoric Water Line for Yakutsk (dashed line; $\delta$D = 7.59 * $\delta$18O – 6.8), based on own data (monthly mean precipitation values between 1997 and 2006; N=106; from Kloss (2008), is given for comparison, and indicative for more continental climate conditions. "

And also on Line 1098: "Additionally, as lacustrine $\delta$18Odiatom also reflects the isotopic composition of the water where the diatoms grow ($\delta$18Olake), 18Odiatom typically reflects meteoric inputs associated with precipitation and riverine inflows (Fig. 6b). For example, existing studies have used lacustrine 18Odiatom to reconstruct past changes in precipitation amount and seasonality, the precipitation/evaporation balance, spring snow melt inputs, and synoptic-scale shifts in atmospheric circulation (Bailey et al., 2015;Meyer et al., 2015;Bailey et al., 2018;Kostrova et al., 2013;Mackay et al., 2013). It is envisaged that changes in 18Odiatom through time at a single site in Bolshoe Toko will yield insights into the long-term air temperature and paleohydrological history of the region."

*And I would suggest dissolution (Smith et al. 2016), which I'll come back to below.*

We agree. Please see our answer to the detailed comment below (Line 976-977).

*Lines 147-151: great to see testable hypotheses here, but they are very vague. As phrased, you are almost certainly to find something that correlates, and we kind of know already from many studies that habitat properties will play a role in influencing bioindicators (e.g. coring from shallow or deep parts of lakes). It might be more useful to hypothesize about Lake Bolshoe Toko specifically?*

We agree and reworded our hypotheses accordingly, reduced to one single sentence that is better suited for our case study: "Bioindicators and abiotic sediment properties will respond to different habitat conditions and lake zonation including water depth, proximity to the main inflow in the South and old moraines in the North of lake Bolshoe Toko."

*Line 163: which is meant by a stressor in a non-impacted lake?*

To simplify we reworded to: "(3) to attribute proxy variability to environmental factors".

*Study Site:*
*Line 190-191: Would be useful to give recent productivity figures*

We agree in principle. Unfortunately, we, including our Russian colleagues from Yakutsk, were not able to access recent numbers of the industrial productivity in this very remote region of Siberia.

*Material and Methods:*
*These are all largely fine. What would be useful to know is (i) why not all 42 core tops were analysed for each proxy, ie give a rationale for looking at numbers of samples detailed for each indicator; (ii) with each proxy, state why it is being measured. For example, what are 18Odiatom values expected to reveal about the environment? Otherwise, there is the danger that the study becomes a bit descriptive.*

We agree. Basically, the numbers of samples vary due to either the amount of sample recovered used up by destructive measurements being not available for further analyses, or the work capacity in our laboratory. The first author planned to finance the study by a third party fund, but the project proposal was rejected. Over the last years starting from late 2013 we steadily analyzed the samples on voluntary level. We still believe that the numbers of samples and analyses shown provide

valuable data on the spatial variability of common proxies applied for palaeonvironmental studies in Bolshoe Toko.

Recent d18Odiatoms values in space are valuable data to plan palaeoclimate studies using this proxy in this lake (planned in the next paper) and also in comparable lakes for interpretation of the data and assessment of a general bias caused by the high variability of lake systems.

We reworded parts of our methods descriptions accordingly to explain the purposes of our analyses and improved the discussion part, i.e. the palaeoenvironmental merits.

*Line 270: alkalinity?*

OK, we changed "Hydrogen carbonate concentrations" to "Alkalinity".

*Lines 410-412: dissolution and concentration calculations are not statistical analyses – more to section above.*

We agree and shifted the descriptions of these to the diatom chapter above.

*Line 426: PCA may capture more variance, but do the data have a horseshoe shape when sample scores only are plotted for Axis 1 and 2? DCA is mainly done to get rid of this artefact. If this is present, then PCA is not appropriate.*

Thank you for the careful consideration. We checked this. And it is not the case. Moreover, the PCA for diatoms and chironomids have been done at an early stage of the analysis and was not presented in the manuscript due to the big volume of the manuscript and because we concentrated primarily on the influence of ecological factors on distribution of bioproxy and not on the distribution of the bioproxy itself. Test PCA has been done to identify which of the methods, lineal or unimodal capture more variance of the data. Results of the both, DCA and PCA are presented in the ESM, Tables a and c. Additionally we show now the PCA biplots in the ESM as well to support our decision to use lineal methods.

*Line 442-443: just a comment: given likely collinearity of many of the explanatory variables, p = 0.05 is quite high, and therefore significance easy to achieve. Might be better to consider a more robust p value of e.g. 0.005 or or even 0.001 to determine what is important (Colquhoun 2015).*

To reveal the intercorrelated parameters in the RDA, first we made an analysis using VIF, as it is described in the Methods section (431-443) so that only non-correlated parameters were retained for the further analysis. Yes, we agree, lower p values are

more robust. However, p< 0.05 is frequently used for interpretation of the data in particular when using the Monte Carlo test with 999 permutations in the CANOCO (Self et al., 2011; Eggermont et al., 2006; Bajolle et al., 2018; Lang et al., 2018; Reid et al., 2018; Reavie and Cai, 2019, etc). Therefore, we believe that the performed tests even under p< 0.05 were helpful for understanding the relationship between the diatoms or chironomid distribution and the environmental factors.

Although Distance to the river and presence of Vegetation showed lower than TOC/N and Water Depth level of significance (p value slightly higher than 0.05), but were still used for interpretation of the chironomid data as the next tested parameters (TC, Distance to the shore, Silt, Clay) had much higher p values and were clearly insignificant (0.25+). We document all this carefully in the methods section.

REF.: Bajolle L., Larocque-Tobler I., Gandouin I., Lavoie M., Bergeron Y., et al.. 2018. Major postglacial summer temperature changes in the central coniferous boreal forest of Quebec (Canada) inferred using chironomid assemblages. Journal of Quaternary Science, 33 (4), pp.409 - 420. DOI: 10.1002/jqs.3022. hal-01890700

Lang B., Medeiros A. S., Worsley A., Bedford A., Brooks S.J. 2018. Influence of industrial activity and pollutionon the paleoclimate reconstruction from a eutrophic lakein lowland England, UK. J Paleolimnol, 59:397–410 https://doi.org/10.1007/s10933-017-9995-6

Reavie E.D., Cai M. 2019. Consideration of species-specific diatom indicators of anthropogenic stress in the Great Lakes. PLOS ONE14(5): e0210927. https://doi.org/10.1371/journal.pone.0210927

Reid M.A., Chilcott S., Thoms M.C. 2018. Using palaeoecological records to disentangle the effects of multiple stressors on floodplain wetlandsJ Paleolimnol. 60:247–271 https://doi.org/10.1007/s10933-017-0011-y

Self, A.E., Brooks, S.J., Birks, H.J.B., Nazarova, L., Porinchu, D., Odland, A., Yang, H., Jones, V.J. 2011. The distribution and abundance of chironomids in high-latitude Eurasian lakes with respect to temperature and continentality: development and application of new chironomid-based climate inference models in northern Russia. Quaternary Science Reviews 30, 1122-1141. (already cited)

Eggermont H., Heiri O., Verschuren D. 2006. Fossil Chironomidae (Insecta: Diptera) as quantitative indicators of past salinity in African lakes. Quaternary Science Reviews 25: 1966–1994.

*Line 452: define Hill's N2 here*

We agree, and defined. It is known that less reliability should be placed on the samples in which more than 5% of the taxa are not represented in the modern

calibration data or more than 5% of the taxa are rare in the modern calibration dataset (i.e., if the effective number of occurrences in the training set, the Hill's N2, are less than 5) (Hill, 1973; Heiri and Lotter, 2001;Self et al., 2011).

Ref.: Hill MO. 1973. Diversity and evenness: a unifying notation and its consequences. Ecology 54:427–432.

*Line 460-461 – is this a form of PCA?*

The endmember modelling algorithm by Dietze et al 2011 uses eigenspace analyses with a combination of known scaling procedures and multivariate methods including factor analyses and PCA. The resulting loadings allow the respective endmembers (mainly grain-size distributions originated by sediment transport processes) to be characterized, while the scores indicate the proportions of variance of each endmember within a sample.

*Line 465: given likely very strong autocorrelation in the spatial datasets, p = 0.05 is quite high, and therefore significance easy to achieve. Might be better to consider a more robust p value of e.g. 0.005 or or even 0.001 to determine what is important (Colquhoun 2015). Or test significance once spatial autocorrelation has been taken into account.*

We agree and modified our analyses accordingly. We applied spatial autocorrelation analysis using coordinates of sample sites in R package "ape", Moran's I autocorrelation coefficient displayed as p values. This analysis is now part of the correlation figure as part b. We additionally refined the Pearson matrix (part a) with more severe p values setting 0.001 as limit to assign significant correlations. We added the description of the spatial autocorrelation test in our methods and changed the text in the figure caption. The spatial autocorrelation enabled us also to state more environmental interpretations in the discussion section. Thank you very much for this valuable hint.

We added to the discussion: "High autocorrelation coefficients (Moran's I p values) for species richness and valve concentration indicate strong local influence of biotic processes, i.e. reproduction, leading to spatial autocorrelation (Legendre et al., 2005). The lowest observed autocorrelation for the diatom planktonic/benthic ratio confirms the strong relationship between diatom species assemblage composition and water depth." Comment: We believe that this is true because the nature of water depth data in a lake suggest lowest possible spatial autocorrelation values as well.

*Fig 2 & 3: It really would be better to show the distribution of the biological proxies using indirect ordination first of all, and then show the more advanced ordination (RDA).*
*A biplot of samples ordered by indirect ordination, just on the basis of biological composition, can be immensely informative.*

> We agree. Do to the large dimension of the manuscript we now added the PCA biplots of diatoms and chironomids in the Electronic Supplementary Material.

*In the RDA, did you test the explanatory variables for normality before analyses? Which ones had to be transformed before the analyses? I can't see this information anywhere, yet this is very important.*

> Yes, this was done as well at the earlier stage of the manuscript preparation.
> We add the information to the Method section (at the beginning of the line 463):
> "All explanatory variables were tested for normality prior to the analyses. Skewness reflects the degree of asymmetry of a distribution around its mean. Normal distributions produce a skewness statistic of about zero. Values that exceeded 2 standard errors of skewness were identified as significantly skewed (Sokal and Rohlf, 1995). Environmental variables with skewed distributions (gravel, grain-size EM2, smectite-chlorite, mica, K-feldspar) were log transformed and remaining parameters were left untransformed. To reveal intercorrelated parameters, we performed a variance inflation factor (VIF) analysis prior to ordination techniques to only retain non-correlated parameters in further multivariate analysis."
> REF: Sokal, R.R., Rohlf, F.J., 1995. Biometry: The Principles and Practice of Statistics in Biological Research. W. H. Freeman and Co, New York.

*Fig 5: In one sense these analyses are fine. But as autocorrelation will be so high here, I think a more robust p values is warranted, else there is a danger of getting lots of Type 1 errors*

> We cite here our answer to your comment on *Line 465:* We agree and modified our analyses accordingly. We applied spatial autocorrelation analysis using coordinates of sample sites in R package "ape", Moran's I autocorrelation coefficient displayed as p values. This analysis is now part of the correlation figure as part b. We additionally refined the Pearson matrix (part a) with more severe p values setting 0.001 as limit to assign significant correlations. We added the description of the spatial autocorrelation test in our methods and changed the text in the figure caption. The spatial autocorrelation enabled us also to state more environmental interpretations in the discussion section. Thank you very much for this valuable hint.

*Line 534-535: Does it matter that virtually none of the samples actually lie on the GMWL? The lagoon shows signs of evaporation, but almost all the samples are below the GMWL, so are the isotopes influenced by ocean sources with high humidity? Results: Table 1: It might be useful to show these relationships in a PCA, with variables standardised to take account of different units... Only include variables above detection limits.*

We have now used monthly mean precipitation values between 1997 and 2006 from Yakutsk (Kloss, 2008) to construct a local meteoric water line (LMWL) for comparison to our lake isotope data. The LMWL (Yakutsk) has been added to Figure 6b, and plots slightly below the GMWL. The Bolshoe Toko water isotope samples lie in between the GMWL and the LMWL, but closer to the prior. This demonstrates that a meteoric origin is possible if not likely. The slighty offset from the GMWL (and a d excess < 10‰) is typical for Pacific-bound moisture and has also been found elsewhere i.e. in Kamchatka (Meyer et al., 2015 GPC). Taking these arguments into account, the manuscript text has been revised accordingly:

"All samples are positioned close to the Global Meteoric Water Line (GMWL, Fig. 6) indicating negligible evaporative effects on lake water isotope composition, and a dominant influence of meteoric inputs both directly (i.e., precipitation) and indirectly (i.e., river inflows). The Local Meteoric Water Line for Yakutsk (dashed line; dD = 7.59 * d18O − 6.8), based on own data (monthly mean precipitation values between 1997 and 2006; n=106; from Kloss, 2008), is given for comparison, even though Yakutsk is characterised by more continental climate conditions than BT".

While we agree that a PCA would also be interesting; further investigating all the potential environment controls on lake water isotopes (e.g. moisture sources, transport, humidity etc.) is really beyond the scope of this paper.

*Line 637: What does this index tell us?*

OK, we added: "index for high nutrients and low pH (Smol et al., 1984)". This index is explained in the discussions afterwards as well.

*Fig 7- 10 With all of these figures, I'm not convinced by the use of the green - red scale to represent low to high; how are scales chosen? Why do some maps have purple? It would be good to know how objective the choice of scales was.*

We agree that we should have explained better. We now added the explanation as follows in the caption of all maps "Maps compiled in ArcGIS 10.4. Scales chosen as 10 classes with equal intervals."; and in the caption of Fig. 10 for explaining the

purple part: "Maps e and h had exceptionally high values of achnanthoid and cymbelloid taxa only in the very shallow (0.5 m) site PG2142. These values are shown in purple, indicated separately at the right side of the scales".

Comment: If we would not use the extra purple colour here, but stick to the equal intervals classes (linear scaling), the finer variations between other sites would be damped. Equal interval is, however, a straight forward scaling we would like to keep constantly in all maps.

*Line 929: Can these two be related like this (Simpson diversity – beta diversity)*

This was actually not pointed out in the detailed comments, but only in the pdf comments - we think this is a valuable point worthy to bring up here. As pointed out in the iNEXT description, Hill numbers include the three most widely used species diversity measures as special cases: species richness (q=0), Shannon diversity (q=1) and Simpson diversity (q=2). It is further known (Hill 1973) that Hill's N2 = reciprocal of Simpson diversity. iNEXT further states that their estimated (sample-size based rarefaction) Simpson diversity can be treated as the effective number of dominant species in the assemblage. Furthermore, beta diversity is the species turn over (or simply the rate of change) between diversities in different samples. We therefore agree with your concern and revise our sentence to a more clear statement "the Simpson diversity index suggests higher effective numbers of dominant species associated to increased habitat complexity". Thank you for bringing this up.

*Line 994-995: If this is the rationale for including Chrysophytes, then perhaps state this earlier. Are there any conclusions from their distribution in the lake?*

*Mallomonas* is an easy to identify, well preserved remain of gold-brown algae that is phylogenetically not very well understood. Anderson 1987 suggested that biochemical and ultrastructural features separates this group from Chrysophyceae, and suggested a new class (Synurophyceae). In any case we suppose that the downcore variability of both Chrysophyte cysts and *Mallomonas* can give valuable insights in the past hydrological development at core positions. Chrysophyte cysts were described contradictorily in the literature, as we listed in the manuscript, and *Mallomonas* has generally only very few environmental assumptions available from the literature. Both indices show rather high values near the Utuk river (C:D index revealed significant (p 0.001) negative correlation (r -0.27) with "distance from river". Even though these values are vague, these indices can potentially be considered in downcore records as additional proxies for riverine activity. We

added: "…slight tendency towards proximity to river input and high water depths."
In the discussions.

*Line 976-977: the authors should also consider the role that dissolution can play here, especially for younger material, e.g. see Smith et al. 2016. DOI: 10.1002/rcm.7446*

> We agree that dissolution may play a role and might have an impact on the oxygen isotope composition of diatoms especially in younger sediments. This is suggested in the excellent paper of Smith et al. 2016. Since in our study we are doing a comparison of (sub)recent spatially distributed surface samples, the time for dissolution in nature is relatively short (annual to decadal scale) and in this framework more or less identical for all samples. On the other hand, our samples have been prepared with caution at low temperatures as the chemical treatment with peroxide (removal of organics) and hydrochloric acid (dissolution of carbonates) might also have an impact on the isotope composition, especially if peroxide is used at higher temperatures. Moreover, both together (cautious wet chemistry and young sediments) yield a clean diatom fraction of the lowest dissolution type (i.e. DDI 1) indicative of no/low impact of dissolution in nature and during sample preparation.
>
> We therefore just added the following sentences to the text:
>
> "Furthermore, dissolution effects in nature and during sample preparation could have had an impact on $\delta^{18}O_{diatom}$ (Smith et al., 2016). However, we suppose differential dissolution to have minor influence on the spatial variability of $\delta^{18}O_{diatom}$ at BT samples tackled in our study as these are (1) of similar age, (2) have been treated with wet chemistry at low temperatures and (3) after preparation do not show any microscopical signs of dissolution effects (i.e. a low diatom dissolution index, Smith et al., 2016)."

*Line 1061-1062: This need not be the case. For example, in Lake Baikal, Aulacoseira species do very well under the ice, and I'm sure this could be the same for other nonshallow lakes. Eg see Jewson et al. 2009*

> We agree and reworded as follows: "For instance, planktonic communities in Lake Baikal, including *Aulacoseira* species, are found to grow under the ice if the surface snow properties (i.e. thickness, density) allow sufficient light penetration (Jewson et al., 2009;Mackay et al., 2005). Generally, planktonic and benthic diatom species have strategies to survive in ice-covered lakes, for instance growing in benthic

mode, forming resting spores, or attaching to the ice-cover substrate (D'souza, 2012). Hence, the duration and presence of ice-cover can significantly impact both changes in assemblage composition and spatial distribution, particularly including the ratio of planktonic to benthic diatoms (Wang et al., 2012a; Bailey et al., 2018)." Comment: We plan to perform a satellite based study of the lake's history taking into account changes of lake ice duration and spatial structures of first ice break-up and longest ice cover in a next paper.

*Sections 5.4 and the conclusions are all good, but it might be also good to provide a concluding statement about potential for optimal coring location*

We agree and added: "The observed proxy variabilities in the surface sediments suggest at least two locations for sediment coring: (1) at intermediate depth in the northern basin to account for representative bioindicator distributions, and (2) the deep part in the central basin to potentially receive higher temporal resolution in the sedimentary record."

*References used in the review:*
*Colquhoun, D. (2015) An investigation of the false discovery rate*
*and the misinterpreptation of p-values. Royal Society Open Science.*
*http://rsos.royalsocietypublishing.org/content/1/3/140216*
*Jewson, D.H., Granin, N.G., Zhdarnov, A.A., Gnatovsky, R.Y. (2009) Effect of snow*
*depth on under-ice irradiance and growth of Aulacoseira baicalensis in Lake Baikal.*
*Aquatic Ecology, 43, 673–679.*
*Smith, A.C., Leng, M.J., Swann, G.E.A., Barker, P., Mackay, A.W., Ryves, D.B., Sloane,*
*H., Chenery, S.R.N., Hems, M. (2016) An experiment to assess the effects of diatom*
*dissolution on oxygen isotope ratios. Rapid Communications of Mass Spectrometry*
*30, 293-300. DOI: 10.1002/rcm.7446*
*Zhao, Y., Sayer, C.D., Birks, H.H., Peglar, S.M. & Hughes, M. (2006) Spatial representation of*
*aquatic vegetation by macrofossil and pollen remains in a small and shallow*
*lake. Journal of Paleolimnology 35, 335-350.*

*Thank you very much for providing additional literature. We used (and cited) the papers to implement your comments carefully in our manuscript.*

*Please also note the supplement to this comment:*
*https://www.biogeosciences-discuss.net/bg-2019-146/bg-2019-146-RC2-*

[Figure]

[Figure]

*supplement.pdf*

**Further changes marked**

We carefully fine-tuned the content of the paper and performed English proof reading again after the revision. We also highlighted all changes we did beyond the comments of the Reviewers.

---

## Referee Report (RR1)

**Review of Biskaborn et al. revisions submitted to Biogeosciences**

**General comments**

This is a much improved version where the authors have comprehensively addressed reviewer comments from both myself and Emilie Saulnier-Talbot. The manuscript is almost ready for publication, but the authors should take account of the minor amendments suggested below.

**Specific Comments:**

The final line of the abstract, now begs the question: have you taken long cores from either of these two regions as part of your overall long-term studies? This does not need to be considered here, but will need to form part of consideration for site selection in future papers.

**Introduction:**

Line 89: …isotopes in diatom **silica**…

Line 144 and elsewhere: sometimes you use paleo, sometimes palaeo; best to stick with one or other

Fig 1: In the legend, should "drawned" be "drowned"

**Material and Methods:**

Line 273: I think if you want to use AD/BC, I'd recommend using instead CE (common era)

Lines 415-416: Does it matter than New et al. 2002 is quite old now, and the region has seen rapid warming since 1998?

Fig 2a and Fig 3a: as these are diatom and chironomid codes, need to make a link to either species names in the Supp Info, or provide species names alongside codes here in the legend.

**Results:**

Line 526 and elsewhere: ideally for isotopes, use en-dash instead of a hyphen to signify a negative value

**Supplementary Info:**

Fig II: is the y-axis label here correct? it suggests that species are plotted according to water depth; as each site will have a different water depth, I assume that each row is a different site, as indicated in the legend. But labels should be consistent between figure and legend

---

## Author Response (AR2)

**Revision Notes to manuscript bg-2019-146:** *Spatial distribution of environmental indicators in surface sediments of Lake Bolshoe Toko, Yakutia, Russia*

*12 September 2019,* Biskaborn et al.

**Associate Editor Decision: Publish subject to minor revisions (review by editor) (10 Sep 2019) by S. Wajih A. Naqvi**

Comments to the Author:

Dear Dr. Biskaborn

Your revised manuscript has now been seen by one of the original referees. S/he is generally happy with the revision but has still made a few more comments. I request you to kindly consider them carefully and make changes in the manuscript accordingly. I will then examine it myself and we will hopefully complete the review process soon.

Looking forward to hearing from you and with kind regards

Sincerely

Wajih Naqvi

> (Our answers indented and marked in blue)
>
> **Dear Dr. Naqvi,**
>
> Thank you very much for the positive answer and the possibility to resubmit our manuscript after minor revision. We revised the manuscript and the ESM carefully following all comments of the reviewer. We documented all changes in the revision notes provided in the following point-to-point answers.
>
> With best regards on behalf of all authors,
> Boris Biskaborn

[Figure]

[Figure]

**Reviewer Anson Mackay and our answers**

*General comments*
*This is a much improved version where the authors have comprehensively addressed reviewer comments from both myself and Emilie Saulnier-Talbot. The manuscript is almost ready for publication, but the authors should take account of the minor amendments suggested below.*

> **Dear Dr Mackay**
>
> Thank you very much for your repeated voluntary efforts to review our manuscript. We agree with your suggestions and revised the manuscript and the ESM accordingly. We prepared a point-to-point answer to each of your comments below.
>
> With best regards,
> Boris Biskaborn

*Specific Comments:*
*The final line of the abstract, now begs the question: have you taken long cores from either of these two regions as part of your overall long-term studies? This does not need to be considered here, but will need to form part of consideration for site selection in future papers.*

> *Yes, we already retrieved several long sediment cores from Bolshoe Toko, including these two areas with different sedimentological regimes.*
> *We modified the last sentence in the abstract: "Our analyses suggest multiple coring locations preferably at intermediate depth in the northern basin and the deep part in the central basin, to account for representative bioindicator distributions and higher temporal resolution, respectively."*
> *We also indicated the existence of long core material in the method section's field work part: "During this expedition also long core material was retrieved from multiple sites including the northern and central part of the lake and is planned for publication in a separate manuscript."*

*Introduction:*
*Line 89: …isotopes in diatom silica…*
> *Yes, we agree and changed accordingly.*

[Figure]

*Line 144 and elsewhere: sometimes you use paleo, sometimes palaeo; best to stick with one or other*

> *Yes, we agree and changed everywhere in the manuscript to "palaeo".*

*Fig 1: In the legend, should "drawned" be "drowned"*

> *Yes, we agree and revised the figure accordingly.*

*Material and Methods:*

*Line 273: I think if you want to use AD/BC, I'd recommend using instead CE (common era)*

> *Yes, we agree and removed "AD", because it is not necessary there.*

*Lines 415-416: Does it matter than New et al. 2002 is quite old now, and the region has seen rapid warming since 1998?*

> *The New et al. 2002 data set was used for development of the published chironomid based T July inference model (Nazarova et al., 2015) and till now remains one of the most reliable source of information available freely online http://wcatlas.iwmi.org/ where the last version of the data are data to 2009, but the recommended citation is still New at al., 2002. Our very recent comparison (personal, unpublished) of the data provided by http://wcatlas.iwmi.org/ with the https://crudata.uea.ac.uk/cru/data/crutem/ge/ data demonstrated, as expected, nearly no difference. However, we plan to use the more modern data for the development of the next generation of the chironomid-based T July inference model in upcoming papers.*

*Fig 2a and Fig 3a: as these are diatom and chironomid codes, need to make a link to either species names in the Supp Info, or provide species names alongside codes here in the legend.*

> *Yes, we agree and provide the species names alongside the codes in the caption of the figures for the diatom and the chironomid graphs. Please also note: we found that Pliocaenicus bolshetokoensis was not visible in the diatom graph, due to some problem with another label placement, which is now fixed in the revised version.*

*Results:*

*Line 526 and elsewhere: ideally for isotopes, use en-dash instead of a hyphen to signify a negative value*

> *Yes, we agree and changed the symbol. It would be very nice to have additional type setting assistance from the journals type setter, also related to the format of the citations and avoid Word specific automatic formats that we cannot easily control.*

*Supplementary Info:*

*Fig II: is the y-axis label here correct? it suggests that species are plotted according to water depth; as each site will have a different water depth, I assume that each row is a different site, as indicated in the legend. But labels should be consistent between figure and legend*

> *Yes, we agree. We added the sample ID's next to each observation of the relative abundances to the right side of the graph. Now the graph includes both information on the water depth and on the sample ID for comparison with the map provided in figure 1 in the manuscript. (that was actually already done in the first revision effort but the figure was not properly exported from the software, we double checked now, thank you very much for noticing!)*

**Further changes marked**

We carefully fine-tuned the content of the paper and performed English proof reading again after the revision. We also highlighted all changes we did beyond the comments of the Reviewers.